# Irrelevant Context Helps: Understanding the Impact of Context in Large Language Models

## Abstract

Existing context management approaches assume question-irrelevant historical context is detrimental, overlooking the possibility that seemingly unrelated conversations may provide cognitive activation benefits similar to those observed in human problem-solving. We introduce "contextual utility prediction" to address this gap and present When2Read—a benchmark comprising 31,900 comparison pairs across eight domains with systematic manipulation of context dimensions including task consistency, multi-turn depth, question count, and difficulty order. Evaluating eight state-of-the-art LLMs, we demonstrate that the impact of question-irrelevant historical context differs across tasks and settings. In some cases it improves model performance (up to **18.6%**), while in others it results in performance drops (as much as **7.3%**). These bidirectional effects indicate that question-irrelevant context is neither consistently beneficial nor consistently harmful, but highly situation-dependent, leaving no simple rule for when to use it. Inspired by insights from cognitive science, we use output length as a heuristic indicator: models that generate shorter responses than their no-context baseline tend to perform poorly in over 90% of cases, offering a training-free criterion for context selection. Our findings challenge the assumption that question-irrelevant context universally harms performance and offer a practical solution for adaptive context management without model retraining. This work establishes a foundation for dialogue systems that strategically leverage diverse conversational histories.

## 1 Introduction

Large language models (LLMs) have achieved remarkable progress across diverse tasks, driving their deployment in real-world applications including personal assistants, search engines, and multi-turn dialogue systems (Li et al., 2024; Jin et al., 2025; Yi et al., 2025). In these applications, users frequently shift from one topic to completely unrelated ones, e.g., from weather questions to programming tasks, where the weather conversation becomes what we term question-irrelevant context (Laban et al., 2025). To preserve response quality, existing studies primarily rely on filtering out such question-irrelevant context as noise that potentially degrades performance (Wu et al., 2024; Carraro & Bridge, 2024; Zhang et al., 2023). However, this paradigm has not systematically examined whether question-irrelevant context might actually be beneficial under certain conditions.

Addressing this question, cognitive science (Kounios & Beeman, 2014; Benedek & Fink, 2019) provides important insights: humans often gain inspiration from seemingly unrelated experiences when solving problems, enhancing performance by activating reasoning patterns. Inspired by this, we propose the hypothesis that question-irrelevant context may not only constitute noise, but could actually assist LLMs in task completion under specific conditions. As shown in Figure 1, irrelevant content can prompt the model to generate more detailed and comprehensive outputs. Based on these observations, we propose the following three research questions:

- *RQ1:* How can we systematically measure the impact of question-irrelevant context?
- *RQ2:* When does question-irrelevant context improve or decrease LLM performance?
- *RQ3:* Can we leverage question-irrelevant context to systematically improve LLM outputs?

To answer RQ1, we propose the *When2Read Benchmark*, the first large-scale diagnostic benchmark for systematically evaluating the improvement of question-irrelevant context on LLMs. This benchmark

Figure 1: Two illustrative examples showing that question-irrelevant context can improve model explanation and accuracy, challenging the assumption that irrelevant context only introduces noise.

evaluate 8 different LLMs covers 8 representative task domains, including writing, reasoning, and coding, comprising a total of **31,900** question pairs. Each pair consists of a version with question-irrelevant context and a version without, enabling to quantify the impact of question-irrelevant context through rigorous paired comparisons. In the with context setting, question-irrelevant context is systematically compared along four dimensions: task consistency, multi-turn depth, question count, and difficulty order. These comparison are based on validated conversational interaction patterns (Yeomans et al., 2023; Casillas & Frank, 2017), ensuring that the benchmark captures realistic dialogue structures. To ensure the quality of the question-irrelevant context and evaluation questions, we constructed the context from existing real-world datasets and conducted human validation of the generated dialogues to guarantee their reliability. Model outputs are evaluated along two dimensions: accuracy and task-specific quality. This dual-layered design provides a comprehensive and systematic measurement of how question-irrelevant context affects LLM performance.

To answer RQ2, we evaluate LLMs across task domains, context dimensions, and model types (Section 4). Performance is measured by the pairwise win rate—the proportion of cases where responses with question-irrelevant context outperform those without, under correctness and task-specific rubrics. The results reveal clear heterogeneity: the impact of such context can go in both directions depending on the task and setting. In some cases, performance improves substantially (up to 18.6%), while in others it drops noticeably (as much as 7.3%). These bidirectional effects indicate that question-irrelevant context is neither consistently beneficial nor consistently harmful, but highly situation-dependent, leaving no simple rule for when to use it.

To answer RQ3, we examine how models can systematically leverage question-irrelevant context (section 5). Inspired on cognitive science theories of linguistic length effects, we discover that output length serves as a reliable performance indicator: when models produce shorter outputs with context than without, performance degrades in 95% of cases. This simple heuristic achieves up to 31.6% accuracy improvements over generic approaches like COT on choosing the right context.

In summary, our contributions are as follows:

- We introduce When2Read, the first large-scale benchmark for systematically evaluating when question-irrelevant context helps or hurts LLM performance, comprising 31,900 comparison pairs across eight diverse domains with controlled manipulation of context dimensions.
- Our extensive evaluation reveals that question-irrelevant context exhibits highly situation-dependent effects, indicating that it is neither inherently beneficial nor harmful and that no simple rule exists for deciding when to include it.
- We identify a simple yet effective behavioral signal for context selection: when models produce shorter outputs with context than without, performance degrades in 95% of cases. This training-free heuristic achieves up to 31.6% performance improvements over baseline approaches.

## 2 PROBLEM FORMULATION

In multi-turn dialogue systems, a fundamental challenge emerges: deciding when question-irrelevant context improves model performance versus when it instead introduces detrimental noise. We

formalize this as a question-irrelevant context utility prediction problem under the framework of sequential decision-making for large language models.

Let $\mathcal{D} = \{(H_i, q_i, y_i)\}_{i=1}^N$ represent a dataset of multi-turn dialogue instances, where $H_i = \{(u_1, r_1), (u_2, r_2), \ldots, (u_{t-1}, r_{t-1})\}$ denotes the historical context consisting of $t-1$ user-assistant interaction pairs, $q_i$ represents the current user query at turn $t$, and $y_i$ is the ground truth response or evaluation target. Denote $f_\theta$ as an LLM with parameters $\theta$. For any given query $q_i$, we consider two response generation scenarios:

$$\hat{y}_i^{\text{context}} = f_\theta(q_i \mid H_i) \tag{1}$$

$$\hat{y}_i^{\text{no-context}} = f_\theta(q_i). \tag{2}$$

**Question-irrelevant Context Utility Formulation** We define the *question-irrelevant context utility* $U(H_i, q_i)$ as the performance differential between with context and without responses:

$$U(H_i, q_i) = \mathcal{E}(\hat{y}_i^{\text{context}}, y_i) - \mathcal{E}(\hat{y}_i^{\text{no-context}}, y_i), \tag{3}$$

where $\mathcal{E}(\cdot, \cdot)$ is a task-specific evaluation function measuring response quality. Positive utility indicates beneficial context, while negative utility suggests harmful interference.

**Multi-Dimensional Context Characterization** To systematically analyze question-irrelevant context utility, we parameterize historical contexts along four key dimensions:

- **Task Consistency** $\tau \in \{\text{same}, \text{cross}\}$: Whether historical queries belong to the same task domain as the current query
- **Multi-turn Depth** $d \in \{1, 3, 6, 9, 12\}$: Measures how *deep* the conversation goes for a single seed question—given one seed question, multiple follow-up questions are generated, each dependent on the original question
- **Question Count** $c \in \{1, 2, 3, 4, 5, 6\}$: Captures the *breadth* of historical contexts by varying the number of distinct seed questions, each with a few follow-ups, to examine how multiple independent contexts affect the target question
- **Difficulty Order** $o \in \{\text{easy-hard}, \text{hard-easy}\}$: The progression of question difficulty in the historical sequence

Together, this yields a context characterization function $\phi(H_i) = (\tau, d, c, o)$.

## 3 BENCHMARK DESIGN

To systematically evaluate the question-irrelevant context utility function $U(H_i, q_i)$ defined in Section 2, we construct the WHEN2READ benchmark. As illustrated in Figure 2, When2Read Benchmark covers eight representative tasks and systematically manipulates three task-oriented dimensions under controlled historical settings. This design enables a principled analysis of model behavior across diverse conversational scenarios.

### 3.1 TASK COVERAGE AND DATA CONSTRUCTION

We select eight representative domains that cover the major LLM capabilities required in multi-turn dialogue systems while aligning with prior multi-turn benchmark (Bai et al., 2024; Zheng et al., 2023; Chen et al., 2025a; Joshi et al., 2025; Li et al., 2025b): (1) **Writing** and **Roleplay**: evaluate open-ended generation and

Table 1: Datasets Used in Our Multi-Turn Evaluation.

| Domain | Dataset(s) | Diff. | Ground Truth |
|---|---|---|---|
| Writing | WritingBench (Wu et al., 2025b), LitBench (Fein et al., 2025) | No | No |
| Roleplay | Characteristic AI Agents (Wang et al., 2024b) | No | Yes |
| Reasoning | LogiQA2.0 (Liu et al., 2023) | No | Yes |
| Math | MathBench (Liu et al., 2024), MMLU (Hendrycks et al., 2021) | Yes | Yes |
| Coding | LeetCode Temporal Dataset (Xia et al., 2025) | Yes | Yes |
| Extraction | DocBank (Li et al., 2020) | No | Yes |
| STEM | ARC (AI2 Reasoning Challenge) (Clark et al., 2018) | Yes | Yes |
| Humanities | MMLU (Hendrycks et al., 2021) | No | Yes |

persona consistency; (2) **Extraction**: focuses on information retrieval and factual grounding; (3) **Reasoning** and **Math**: target logical inference and quantitative problem solving; (4) **Coding**: examines program synthesis and functional correctness; (5) **Knowledge I (STEM)** and **Knowledge II (Humanities/Social Science)**: assess domain-specific factual knowledge across diverse disciplines.

From each domain, we sample 100 questions from established benchmarks (detailed in Table 1). These benchmarks provide clear difficulty levels, standardized ground truth answers, and are widely used for comparison, thereby meeting our evaluation requirements. In total, this yields 800 seed

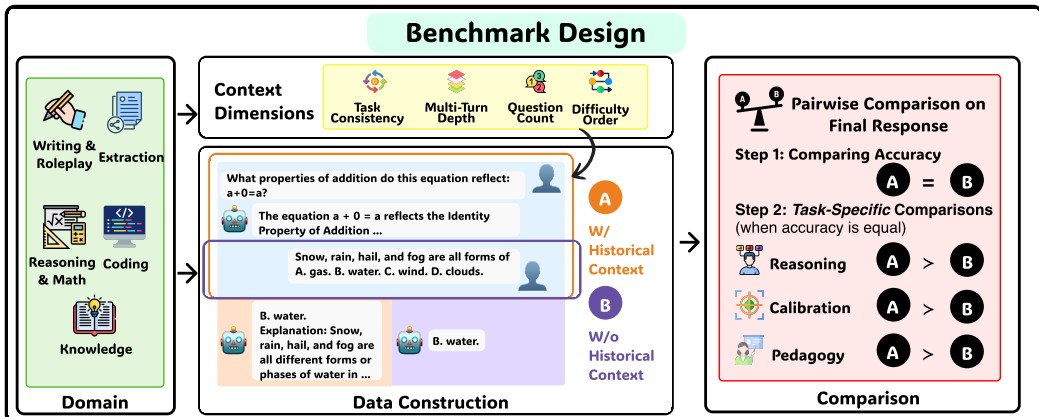

Figure 2: The benchmark spans multiple domains (writing, reasoning, knowledge, coding, and extraction) and systematically manipulates four context construction dimensions (task consistency, multi-turn depth, question count, difficulty order). Data construction includes two conditions: with historical context (multi-turn interactions) versus without historical context (final question only). Evaluation involves pairwise comparison of final responses of conversations with and without historical context, first on accuracy and then—when accuracy is equal—on task-specific qualities (e.g., content, organization, coherence for *writing*).

questions with well-defined ground truth answers. For benchmarks with explicit difficulty levels, such as MathBench and LeetCode, we preserve the original annotations to enable controlled manipulation of difficulty ordering within multi-turn contexts. For open-ended generation tasks such as writing and roleplay, we retain their original settings without imposing artificial difficulty levels, ensuring that the evaluation focuses on output quality and stylistic diversity rather than problem complexity.

## 3.2 Historical Context Construction

The influence of historical context in multi-turn interactions depends not only on whether context is present but also on how it is structured. Real-world conversations exhibit diverse patterns: users may shift topics abruptly, provide long or short histories (Xie et al., 2021), mix easy and difficult questions (Cuayáhuitl et al., 2018), or rely on previous system responses (Deriu et al., 2021). To systematically analyze these factors, we instantiate the context characterization function $\phi(H_i) = (\tau, d, c, o)$ defined in Figure 2 through controlled generation of diverse conversational histories.

Figure 3 illustrates our experimental design for systematically evaluating historical context across four key dimensions. We begin by using GPT-5 to generate complete dialogue sequences from 800 seed questions across eight domains, creating comprehensive multi-turn conversations (as shown in Dimension 1) that serve as the foundation for all subsequent manipulations. Multi-turn Depth (Dimension 1) represents our base dataset: complete dialogues extending up to 12 turns, from which we can truncate to create shorter conversations of 1, 3, 6, or 9 turns. The remaining three dimensions are constructed by strategically sampling and recombining these base dialogues. Question Count (Dimension 2) extracts segments containing 1, 3, or 6 independent questions from the original dialogues. Task Consistency (Dimension 3) controls domain alignment by either maintaining fragments from the same domain or splicing fragments from different domains before presenting the target question. Difficulty Order (Dimension 4) arranges dialogue segments according to difficulty progression (easy-to-difficult or difficult-to-easy) using preserved difficulty annotations. This approach ensures both experimental control and natural dialogue quality.

All original 800-dialogue fragments undergo dual-annotator evaluation for relevance and coherence (rubrics details in Section B.2). The modular design allows us to systematically manipulate specific context characteristics while maintaining conversational coherence—since all experimental conditions derive from the same high-quality dialogue foundation, we can isolate the effects of each dimension.

## 3.3 Evaluation Metrics and Protocols

To directly measure the question-irrelevant context utility function $U(H_i, q_i) = E(\hat{y}_i^{\text{context}}, y_i) - E(\hat{y}_i^{\text{no-context}}, y_i)$ defined in Section 2, we construct controlled comparison pairs from our generated contextual variations. For each seed question $q_i$, we create evaluation pairs comparing model

Figure 3: Overview of our experimental design for evaluating question-irrelevant historical context across four dimensions: (1) Multi-turn Depth (up to 12 turns per dialogue), (2) Question Count (1–6 independent questions), (3) Task Consistency (same vs. different domains), and (4) Difficulty Order (easy-to-difficult vs. difficult-to-easy). These dimensions are visualized in the figure to illustrate how context conditions are systematically constructed for controlled evaluation.

responses under two conditions: the context condition where the model receives the historical context $H_i$ before answering $q_i$, and the no-context condition where the model receives only the target question $q_i$ without any context. Each contextual variation generated in Section 3.2 produces one comparison pair, enabling systematic analysis across all dimensional combinations $\phi(H_i) = (\tau, d, c, o)$. For example, a Math seed question paired with a 6-turn cross-domain history creates one comparison pair, while the same question with a 3-turn same-domain history creates another distinct pair. This paired design isolates the impact of specific contextual configurations, allowing us to determine when historical information helps, hurts, or has no effect on model performance.

We adopt a hierarchical evaluation approach to overcome the limitations of accuracy-only methods, which cannot capture quality differences between responses that are both correct (or both incorrect). Correctness is judged strictly against gold-standard human-verified labels—dataset annotations, LeetCode solutions, and standardized test keys—that we directly inherit from the original sources (Table 1), ensuring exceptionally well-grounded results. When one response is correct and the other is not, the outcome is determined immediately. When both succeed or both fail, we turn to domain-specific quality dimensions (Table 2) for finer insights. Each context–no-context pair is then evaluated dimension by dimension with GPT-5 (validated in 3.4), and the final decision is made through majority voting across dimensions. Aggregating these pairwise results across all domains yields the reported probabilities: values above 0.5 indicate that context helps more often than it hurts, while values below 0.5 suggest the opposite.

The final When2Read benchmark comprises 31,900 comparison pairs systematically distributed across nine domains and all experimental conditions. Each domain contributes proportionally based on the applicable dimensional combinations, ensuring balanced coverage of task consistency, multi-turn depth, question count, and difficulty ordering factors. This extensive scale enables robust statistical analysis of context utility patterns while maintaining sufficient instances within each experimental cell for reliable comparative evaluation across diverse conversational configurations.

## 3.4 EVALUATION VALIDATION

For nuanced evaluation beyond correctness, we rely on LLMs for annotation. Our dataset contains 31,900 comparison pairs from diverse multi-turn interactions, making full human annotation impractical. To scale reliably, we use GPT-5 as a proxy evaluator, validated against three independent human annotators on 350 cases (Appendix C.1). As shown in Figure 8, GPT-5's ratings align closely with humans: Cohen's $\kappa$=0.680, Pearson's $r$=0.839 ($p < 0.001$), exact agreement 0.755, and within-1-point

Table 2: Evaluation dimensions for all tasks. Full rubrics and examples are in Appendix C.1.

| Task | Dimension 1 | Dimension 2 | Dimension 3 |
|---|---|---|---|
| Writing | Content (McKenzie, 2023) | Organization (Yavuz et al., 2025) | Coherence (Li & Huang, 2022) |
| Roleplay | Consistency (Kim et al., 2020) | Immersion (Wang et al., 2025) | Empathy (Chen et al., 2025b) |
| Coding | Robustness (Thakur et al., 2025) | Completeness (Zheng et al., 2024) | Correctness (Zheng et al., 2024) |
| Math | Correctness (Shridhar et al., 2022) | Generalization (Roy & Roth, 2017) | Clarification (Zhang et al., 2022) |
| Reasoning | Grounding (Chandu et al., 2021) | Counterfactuals (Bottou et al., 2013) | Calibration (Wang, 2024) |
| Extraction | Fidelity (Luo et al., 2025) | Specificity (Mavi et al., 2024) | Calibration (Geng et al., 2024) |
| Knowledge I | Reasoning (cau, 2025) | Calibration (Li et al., 2025c) | Pedagogy (Sonkar et al., 2024) |
| Knowledge II | Specificity (Kryjevskaia et al., 2012) | Qualification (Siddique et al., 2025) | Clarity (Pisano et al., 2021) |

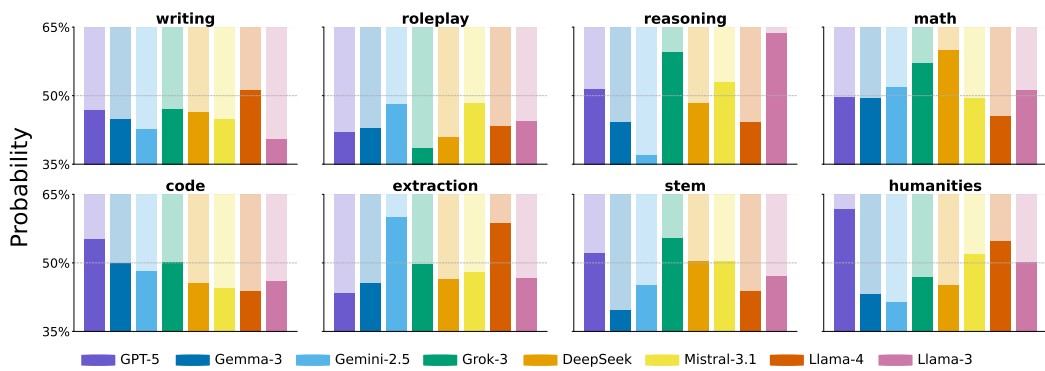

Figure 4: Model Performance with Question-irrelevant Context Across Eight Domains

agreement 0.928. Most disagreements were minor, confirming GPT-5 as a consistent, reliable proxy for assessing follow-up quality in multi-turn dialogue. Further details appear in Appendix C.2.1.

## 4 EXPERIMENTS AND RESULTS

Based on *When2Read Benchmark*, we evaluate eight representative language models through our dual-dimensional analysis framework. We investigate question-irrelevant context effects from two complementary perspectives: **task domain dimensions** and **context dimensions**. The evaluated models include GPT-5 (OpenAI, 2025), Llama-3.1-8B-Instruct (Meta, 2024), Llama-4-Maverick-17B-128E-Instruct (Touvron et al., 2023), Grok-3-mini (xAI, 2025), Gemini-2.5-flash (Google, 2025a), DeepSeek-R1 (DeepSeek-AI, 2025), Gemma-3-4b-it (Google, 2025b), and Mistral-Small-3.1-24B-Instruct (Jiang et al., 2023), across different model families and capability levels.

### 4.1 TASK DOMAIN DIMENSION ANALYSIS

Figure 4 presents the evaluation results from our When2Read Benchmark across eight domains, where performance values indicate the probability that responses with question-irrelevant context outperform those without (values above 0.5 suggest context helps, below 0.5 suggests context hurts performance). The results reveal a notable evolution in how models utilize question-irrelevant context. Earlier models tend to show reduced performance when exposed to question-irrelevant context across most domains, particularly in Writing and Roleplay tasks where scores consistently fall below 0.50. However, state-of-the-art models exhibit a fundamental shift where context no longer uniformly degrades performance, and can even enhance performance substantially in specific domains. For instance, DeepSeek achieves remarkable gains in Mathematical reasoning (0.60), Llama-3 excels in Logical reasoning (0.64), GPT-5 demonstrates superior performance in Humanities (0.62), while Grok-3 shows consistent benefits across multiple quantitative domains (Reasoning: 0.59, Math: 0.57, STEM: 0.55). Notably, advanced models like Gemini (0.60) and Llama-4 (0.59) also exhibit strong improvements in Extraction tasks. **These findings suggest that modern LLMs, like humans in cognitive science studies, can sometimes benefit from question-irrelevant context.** Just as humans derive insights from seemingly unrelated experiences, LLMs may leverage additional context to enhance reasoning and task performance under certain conditions.

### 4.2 CONTEXT DIMENSION ANALYSIS

Figure 5a to 5d investigate four key structural factors that characterize question-irrelevant historical context. **Multi-turn Depth** and **Question Count** show performance improvements in recent models compared to earlier ones. Earlier models like Llama-3 show systematic performance degradation as dialogue turns and question counts increase, while newer models like Llama-4 and GPT-5 maintain stable performance or even demonstrate improvement patterns. **Task Consistency** and **Difficulty Order** expose more nuanced model-specific preferences. Some models demonstrate clear benefits from progressive difficulty increase (easy→difficult), showing improved performance compared to same-difficulty contexts, suggesting they can leverage cognitive scaffolding where simpler problems help build understanding for more complex ones. Conversely, other models like DeepSeek, Grok-3,

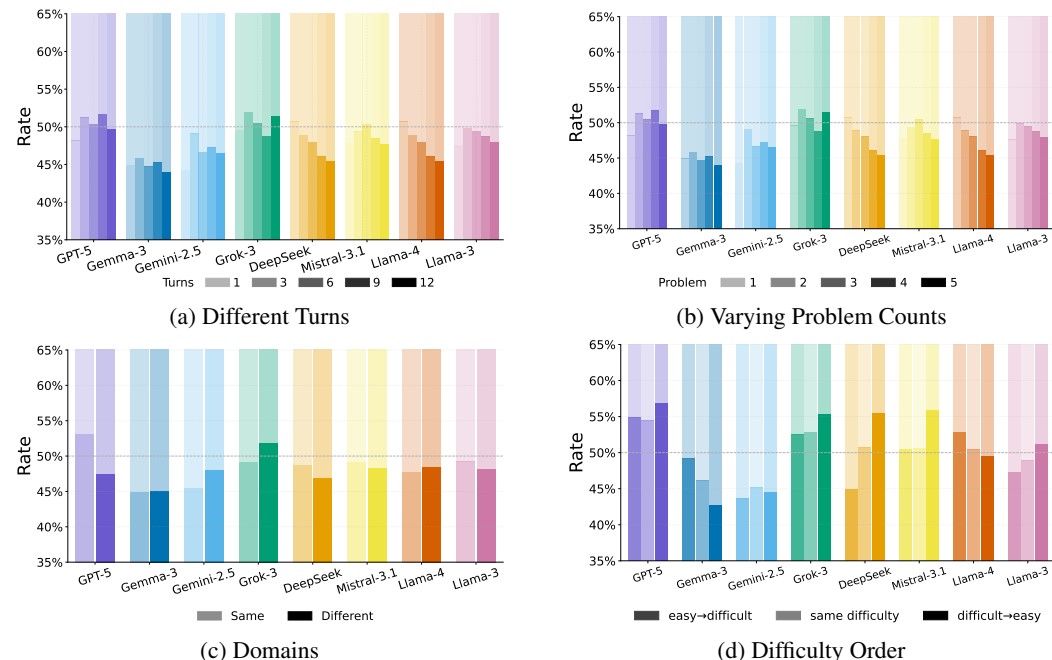

(a) Different Turns

(b) Varying Problem Counts

(c) Domains

(d) Difficulty Order

Figure 5: Question-irrelevant Context across (a) turns, (b) problem counts, (c) domains, and (d) difficulty orders.

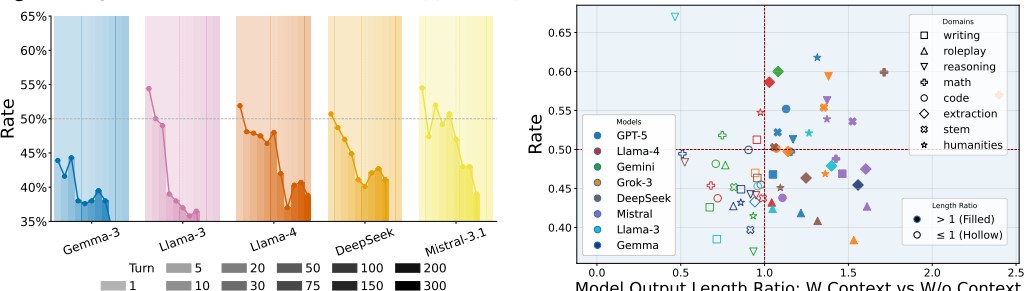

Figure 6: Long-Context Robustness of Open-Source Models across 200 turns

Figure 7: Model Performance vs. Output Length Ratio with Historical Context

and Mistral-3.1 prefer difficult→easy transitions (0.55), indicating they benefit from exposure to complex examples first. **This bidirectional effect reveals the complex role of context, suggesting the need to develop adaptive and context-aware strategies for question-irrelevant context selection rather than relying on fixed rules.**

### 4.3 INSIGHTS AND OPEN QUESTIONS

The results from both the task domain and context dimensions highlight that modern LLMs can, in certain conditions, benefit from question-irrelevant context. **However, this improvement is not uniform: it depends on both the specific task domain and the model's architectural characteristics.** These findings raise two key questions for future investigation: (1) What is the maximum context length over which these performance gains persist? (2) Can we leverage question-irrelevant context to systematically improve LLM outputs?

## 5 ANLYSIS

### 5.1 MAXIMUM CONTEXT LENGTH FOR SUSTAINED PERFORMANCE GAINS

Section 4.2 revealed that modern models can leverage task-irrelevant context to improve performance in moderate-length histories (e.g., 12 turns). To investigate maximum context length for sustained performance gains, we extended the analysis to 300 turns using selected open-source models. Figure 6 shows that models exhibiting improvements in standard ranges eventually experience performance decline—Llama-4 plateaus around 50 turns and starts decreasing beyond 100 turns, while all tested

Table 3: Model performance with prompting and CoT relative to Vanilla.

| Model | Status | Writing | Roleplay | Reasoning | Math | Code | Extraction | STEM | Humanities |
|---|---|---|---|---|---|---|---|---|---|
| GPT-5 | Prompting | 0.51 ↑0.01 | 0.51 ↑0.01 | 0.48 ↓0.02 | 0.49 ↓0.01 | 0.44 ↓0.06 | 0.34 ↓0.16 | 0.48 ↓0.02 | 0.35 ↓0.15 |
| | COT | 0.52 ↑0.02 | 0.52 ↑0.02 | 0.49 ↓0.01 | 0.49 ↓0.01 | 0.44 ↓0.06 | 0.34 ↓0.16 | 0.48 ↓0.02 | 0.35 ↓0.15 |
| | Ours | **0.52** ↑0.02 | **0.55** ↑0.05 | **0.64** ↑0.14 | **0.56** ↑0.06 | **0.60** ↑0.10 | **0.44** ↓0.06 | **0.59** ↑0.09 | **0.54** ↑0.04 |
| Llama-3 | Prompting | 0.54 ↑0.04 | 0.49 ↓0.01 | 0.38 ↓0.12 | 0.49 ↓0.01 | 0.60 ↑0.10 | 0.24 ↓0.26 | 0.47 ↓0.03 | 0.52 ↑0.02 |
| | COT | 0.56 ↑0.06 | 0.50 | 0.39 ↓0.11 | 0.50 | 0.60 ↑0.10 | 0.30 ↓0.20 | 0.47 ↓0.03 | 0.52 ↑0.02 |
| | Ours | **0.57** ↑0.07 | **0.57** ↑0.07 | **0.54** ↑0.04 | **0.51** ↑0.01 | **0.62** ↑0.12 | **0.61** ↑0.11 | 0.47 ↓0.03 | **0.58** ↑0.08 |
| Llama-4 | Prompting | 0.47 ↓0.03 | 0.44 ↓0.06 | 0.57 ↑0.07 | 0.58 ↑0.08 | 0.61 ↑0.11 | 0.04 ↓0.46 | 0.56 ↑0.06 | 0.49 ↓0.01 |
| | COT | 0.47 ↓0.03 | 0.42 ↓0.08 | 0.57 ↑0.07 | 0.58 ↑0.08 | 0.61 ↑0.11 | 0.05 ↓0.45 | 0.56 ↑0.06 | 0.49 ↓0.01 |
| | Ours | **0.58** ↑0.08 | **0.61** ↑0.11 | 0.49 ↓0.01 | **0.64** ↑0.14 | 0.50 | **0.56** ↑0.06 | **0.52** ↑0.02 | **0.57** ↑0.07 |
| Gemini | Prompting | 0.03 ↓0.47 | 0.05 ↓0.45 | 0.14 ↓0.36 | 0.34 ↓0.16 | 0.25 ↓0.25 | 0.02 ↓0.48 | 0.31 ↓0.19 | 0.20 ↓0.30 |
| | COT | 0.45 ↓0.05 | 0.45 ↓0.05 | 0.54 ↑0.04 | 0.42 ↓0.08 | 0.52 ↑0.02 | 0.21 ↓0.29 | 0.46 ↓0.04 | 0.53 ↑0.03 |
| | Ours | **0.65** ↑0.15 | **0.61** ↑0.11 | **0.54** ↑0.04 | **0.60** ↑0.10 | **0.58** ↑0.08 | **0.60** ↑0.10 | **0.55** ↑0.05 | **0.52** ↑0.02 |
| Gemma | Prompting | 0.03 ↓0.47 | 0.05 ↓0.45 | 0.13 ↓0.37 | 0.34 ↓0.16 | 0.25 ↓0.25 | 0.02 ↓0.48 | 0.31 ↓0.19 | 0.20 ↓0.30 |
| | COT | 0.04 ↓0.46 | 0.04 ↓0.46 | 0.13 ↓0.37 | 0.34 ↓0.16 | 0.22 ↓0.28 | 0.02 ↓0.48 | 0.28 ↓0.22 | 0.20 ↓0.30 |
| | Ours | **0.56** ↑0.06 | **0.50** | **0.52** ↑0.02 | **0.57** ↑0.07 | **0.51** ↑0.01 | **0.57** ↑0.07 | **0.52** ↑0.02 | **0.53** ↑0.03 |
| DeepSeek | Prompting | 0.52 ↑0.02 | 0.45 ↓0.05 | 0.53 ↑0.03 | 0.27 ↓0.23 | 0.56 ↑0.06 | 0.25 ↓0.25 | 0.48 ↓0.02 | 0.50 |
| | COT | 0.52 ↑0.02 | 0.42 ↓0.08 | 0.53 ↑0.03 | 0.27 ↓0.23 | 0.58 ↑0.08 | 0.23 ↓0.27 | 0.50 | 0.49 ↓0.01 |
| | Ours | 0.46 ↓0.04 | 0.47 ↓0.03 | 0.47 ↓0.03 | **0.66** ↑0.16 | 0.49 ↓0.01 | 0.45 ↓0.05 | **0.51** ↑0.01 | **0.59** ↑0.09 |
| Grok-3 | Prompting | 0.54 ↑0.04 | **0.64** ↑0.14 | 0.41 ↓0.09 | 0.33 ↓0.17 | 0.44 ↓0.06 | 0.46 ↓0.04 | 0.42 ↓0.08 | 0.55 ↑0.05 |
| | COT | 0.54 ↑0.04 | 0.64 ↑0.14 | 0.41 ↓0.09 | 0.33 ↓0.17 | 0.44 ↓0.06 | 0.46 ↓0.04 | 0.42 ↓0.08 | 0.55 ↑0.05 |
| | Ours | **0.57** ↑0.07 | 0.51 ↑0.01 | **0.50** | **0.70** ↑0.20 | **0.62** ↑0.12 | **0.51** ↑0.01 | **0.54** ↑0.04 | **0.60** ↑0.10 |
| Mistral | Prompting | 0.43 ↓0.07 | 0.40 ↓0.10 | 0.42 ↓0.08 | 0.47 ↓0.03 | 0.50 | 0.21 ↓0.29 | 0.41 ↓0.09 | 0.44 ↓0.06 |
| | COT | 0.45 ↓0.05 | 0.37 ↓0.13 | 0.41 ↓0.09 | 0.45 ↓0.05 | 0.49 ↓0.01 | 0.18 ↓0.32 | 0.44 ↓0.06 | 0.44 ↓0.06 |
| | Ours | **0.53** ↑0.03 | **0.49** ↓0.01 | **0.54** ↑0.04 | **0.55** ↑0.05 | **0.56** ↑0.06 | **0.57** ↑0.07 | **0.52** ↑0.02 | **0.47** ↓0.03 |

models ultimately degrade, albeit with different threshold patterns. These results indicate that performance gains from task-irrelevant context do not persist under extremely long contexts, highlighting the need for more intelligent context management when dealing with very long sequences.

## 5.2 EXPLORING STRATEGIES TO LEVERAGE TASK-IRRELEVANT CONTEXT

To understand why task-irrelevant context benefits model performance, we examine output length through a cognitive lens. Cognitive psychology finds that cognitive activation correlates positively with expressive richness when humans draw inspiration from seemingly irrelevant experiences (Tidikis et al., 2017; Kounios & Beeman, 2014; Benedek & Fink, 2019). We investigate whether large language models exhibit a similar pattern. Our analysis (Figure 7) reveals a striking pattern: models producing shorter outputs than their without-context baseline (length ratio ≤ 1, hollow markers) almost universally exhibit poor performance with win rates below 0.5, suggesting insufficient cognitive activation. Conversely, longer outputs (length ratio above 1, filled markers) show greater variability. Some achieve win rates above 0.5, particularly in reasoning domains, while others remain near baseline. This indicates that extended output provides necessary "cognitive space" for beneficial activation but doesn't guarantee success. These findings suggest a practical strategy: **reject outputs shorter than the no-context baseline, as these likely indicate failed cognitive activation, thereby improving reliability of performance gains from irrelevant context.**

## 5.3 COMPARISON AGAINST STANDARD CONTEXT PROCESSING APPROACHES

Based on Section 5.2, we use the length of the output from the models as a gating criterion, rejecting instances where output length falls below the without context baseline. This design choice accommodates both closed-source and open-source models, as output length is universally accessible regardless of model transparency. To demonstrate the effectiveness of our method, we compare against two famous non-training approaches: standard prompting and chain-of-thought (CoT) prompting (Wei et al., 2023) (implement details in Appendix E). The reported values represent win rates where our method's outputs are compared against the standard context baseline (where all available context is provided to the model, as is typical in current deployments) using pairwise evaluation. Values above 0.5 indicate that our length-filtered outputs are preferred, while values below 0.5 suggest the baseline performs better. Table 3 shows that prompting methods fail to leverage task-irrelevant context effectively. Standard prompting produces consistent degradation across models, with Gemini experiencing severe losses (Writing: 0.39, Reasoning: 0.23) and Gemma showing widespread decline

(Writing: 0.42, Reasoning: 0.31). COT provides little improvement, often replicating the same degradation patterns. In contrast, our simple yet effective strategy of rejecting outputs shorter than the no-context baseline consistently outperforms both baselines, achieving substantial win rates such as Llama-4's 0.58 in Writing and GPT-5's 0.64 in Reasoning. This demonstrates that our length-based filtering provides an effective approach for task-irrelevant context selection.

## 6 RELATED WORK

**Context Selection Paradigm.** Existing approaches to context selection in multi-turn LLM systems operate under a fundamental assumption: relevant context improves performance while irrelevant context degrades it (Zhao et al., 2021; Yi et al., 2025). This paradigm has driven the development of sophisticated selection mechanisms based on semantic similarity (Li et al., 2025a; Ma et al., 2023; Khattab & Zaharia, 2020), and hybrid relevance scoring (Chevalier et al., 2023). These methods actively filter out irrelevant context to keep the conversation coherent. However, this relevance-centric approach inherently overlooks a critical possibility: seemingly irrelevant historical context may provide unexpected benefits through mechanisms beyond direct semantic relevance. Current evaluation frameworks reinforce this limitation by measuring success through coherence and topical relevance metrics (Zhang et al., 2023; Wang et al., 2024a; Laban et al., 2025). This systematic exclusion of 'irrelevant' context reveals a significant blind spot in understanding how historical context actually affects model capabilities.

**Question-irrelevant Context.** Cognitive science research reveals that humans frequently derive insights from seemingly unrelated experiences when solving complex problems (Tidikis et al., 2017; Kounios & Beeman, 2014; Benedek & Fink, 2019). This phenomenon, known as cross-domain knowledge transfer, occurs when exposure to disparate information activates latent reasoning patterns or primes attention mechanisms that enhance problem-solving performance. Studies show that cognitive activation correlates positively with expressive richness during insight generation (Kounios & Beeman, 2014), suggesting that beneficial cognitive states manifest in observable behavioral patterns. These findings challenge the intuitive notion that question-irrelevant context is inherently useless. Despite these insights from human cognition, the potential for analogous cross-domain activation in language models remains largely unexplored.

## 7 CONCLUSIONS, LIMITATIONS AND FUTURE WORK

We introduced When2Read to evaluate when question-irrelevant context helps or hurts LLM performance in multi-turn dialogues. By systematically manipulating context dimensions across diverse domains, When2Read enables fine-grained assessment of context utility under controlled conditions. Our experiments show that question-irrelevant context can systematically improve reasoning performance by up to 18.6% while degrading creative tasks by 7.3%, challenging the assumption that such context is purely detrimental. We introduce output length as a simple behavioral signal that consistently reflects response quality. Leveraging this signal enables training-free context selection, leading to accuracy improvements of up to **31.6%**.

While our study reveal that question-irrelevant context can benefit model performance, it has several limitations. First, our evaluation focuses on dialogue histories up to 300 turns, leaving open questions about ultra-long context settings. Second, our benchmark covers eight models and eight task domains, which, despite being diverse, may not fully represent specialized or multi-modal settings.

Future work can address these limitations along several directions. A promising direction is to develop adaptive context management strategies that dynamically decide when and how much context to read, possibly leveraging uncertainty estimation or reinforcement learning. Extending the benchmark to ultra-long or multi-modal dialogues would improve the robustness and generality of our findings. Another avenue is to build a unified framework for predicting context utility across diverse tasks and models, enabling automated adaptation to new settings. Finally, integrating our insights with retrieval-augmented generation or memory-augmented architectures could lead to systems that selectively exploit both relevant and irrelevant context for improved reasoning and robustness.

## REPRODUCIBILITY STATEMENT

We have made extensive efforts to ensure that our work is reproducible. To support this, we will release and open-source the main dataset of evaluation contexts as well as the code used to generate the multi-turn dataset. This will allow other researchers to scale the dataset to a larger number of turns and examine the scaling effects discussed in this work. The prompts used for data generation are provided in Appendix B.1, while Section 3 describes the seed datasets, algorithms, and problem formulation used in our benchmark. Appendix B.2 details the annotation guidelines and standards, enabling reproduction of our labeling pipeline for those who prefer not to rely solely on our released dataset. The exact metrics and rubric definitions used in LLM evaluation are specified in Appendix C.1, and the naive prompting strategies are included in Appendix E. Finally, Appendix C.2 discloses all model configurations, parameter settings, and experimental environments. Together, these materials are intended to provide full transparency and facilitate reliable reproduction of our experiments.

## ETHICS STATEMENT

Our work relies on previously released datasets as seed material for generating a large-scale multi-turn evaluation benchmark. These datasets span diverse domains including writing, reasoning, coding, STEM, and the humanities, and were explicitly released for research purposes. We selected them because they are high quality, human annotated, and information rich, making them valuable foundations for reproducible and rigorous evaluation. All source datasets are distributed under licenses that permit academic use, redistribution, and adaptation, and we carefully comply with their stated terms.

Our contribution does not involve republishing the original datasets in full. Instead, we systematically extend and transform the seed material to create new multi-turn evaluation items. This transformation serves several ethical purposes: (1) it advances the original intent of the datasets by supporting open scientific research; (2) it is entirely noncommercial, providing a community resource for evaluating dialogue, iterative reasoning, and long context performance; (3) it reshapes the benchmarks into a different evaluation setting focused on multi-turn interactions rather than single-turn tasks; and (4) it complements rather than replaces or competes with the original datasets by broadening their utility.

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

# APPENDIX

# A  LLM USAGE DISCLOSURE

In preparing this manuscript, we used LLMs (ChatGPT-5) solely for editing and proofreading. After the initial draft was written entirely by the authors, the LLM was employed to refine grammar,

enhance clarity, and improve wording. It was not used to generate content, shape research ideas, write code, conduct analyses in place of humans, or contribute substantive scientific text. All scientific contributions, derivations, claims, and experiments are the work of the human authors, who take full responsibility for their accuracy.

This disclosure is provided in accordance with ICLR's policy requiring that any use of LLMs be reported, with authors remaining fully accountable for all content, including text refined with LLM assistance.

## B  DETAILS OF DATASET CONSTRUCTION

### B.1  MULTITURN FOLLOW-UP GENERATION PROMPTS

---

**Generation Prompt**

**Your task is to generate a single, natural follow-up question that a HUMAN USER would most likely ask next, based on the provided conversation.**

**Guidelines for Generating the Question**

1. **Stay in the User's Role**

   - The question must sound like it comes directly from the human user reacting to the assistant's last response.
   - It should express curiosity, clarification, or a request for more detail/examples — not task planning or asking the assistant what it wants to do.

2. **Be Context-Aware**

   - The question should directly connect to the assistant's most recent answer.
   - Assume the user is engaged, interested, and trying to learn more or apply what was just explained.

3. **Tone Matching**

   - Match the tone of the user so far: curious, practical, and conversational.
   - Ask questions that would elicit long responses detailed from the assistant and require more thought than a simple yes/no.

4. **Drive the Conversation**

   - The question should either: a) ask for clarification of something the assistant just said, b) request a deeper explanation/example, or c) explore a natural "next step" from the content provided.

**Hard DO NOT Rules**

- **DO NOT** ask questions from the assistant's perspective (e.g., "Would you like me to add X?").
- **DO NOT** generate meta-questions about the conversation, prompts, or instructions.
- **DO NOT** rephrase the previous user question.
- **DO NOT** propose options for the assistant to choose from.

**Output**

- Only return the single most likely user follow-up question.
- Do not include explanations, meta-comments, or extra formatting.

**Conversation so far:**
{convo_text}

---

| Criterion | Accept Guidance (vs. common Reject cues) |
|---|---|
| Relevance | Directly tied to seed/context; no topic drift or unsupported assumptions. *Reject if*: introduces new, unrelated goals or contradicts given facts. |
| Naturalness | Reads like a plausible human follow-up; appropriate register and specificity. *Reject if*: templated, stilted, or meta-instructions (e.g., "Ask a clarifying question"). |
| Quality/Usefulness | Advances dialogue (clarifies, narrows scope, adds necessary detail). *Reject if*: redundant restatement, trivial yes/no when deeper inquiry needed. |
| **Labeling Rule:** *Accept* requires all three met; any failed criterion $\Rightarrow$ *Reject*; else *Borderline*. | |

Table 4: Operational rubric for human follow-up filtering.

### B.2 Human Follow up and Seed Question Annotation Protocol

To establish a robust human annotation for filtering which turns are relevant and which questions were relevant when constructing our dataset, we implemented a structured annotation protocol. Three independent annotators (with prior experience in discourse analysis and LLM evaluation) rated follow-ups given the seed prompt and the multi-turn dialogue context. Before the main study, annotators completed a calibration phase on 20 pilot samples; disagreements were discussed to harmonize rubric interpretations and finalize decision rules.

**Materials and Blinding.** For each case, annotators received: (i) the seed prompt, (ii) the full preceding dialogue context, and (iii) a single follow-up utterance. Source provenance (human vs. model) was hidden; instances were randomly ordered to mitigate sequence effects.

**Primary Criteria.** Annotators judged each follow-up on three criteria:

- **Relevance:** The follow-up directly pertains to the seed and contextual turns; it does not introduce unrelated topics or hallucinated premises.

- **Naturalness:** The follow-up is phrased as a plausible human question/utterance for the given context (avoids templated or mechanically stilted language).

- **Quality/Usefulness:** The follow-up advances the conversation (e.g., specifies goals, clarifies ambiguity, or deepens inquiry) without redundancy or triviality.

**Decision Labels and Notes.** Each instance received one of three labels:

- **Accept**: Meets all three criteria.

- **Reject**: Fails any one of the criteria (clear violation).

- **Borderline**: Ambiguous or mixed cases; annotators provide a one–two sentence justification (e.g., "technically relevant but awkward phrasing").

**Aggregation and Adjudication.** Annotations were completed independently and aggregated by majority vote. Ties ($1-1-1$ or $1-2$ splits with a *Borderline*) triggered targeted adjudication: annotators briefly reviewed justifications and applied the finalized rubric rules to reach consensus.

**Quality Controls.** We monitored inter-annotator agreement throughout. The calibration phase established minimum expected agreement (Cohen's $\kappa$ threshold) before scaling to the full set; spot-checks of $10\%$ of cases ensured rubric drift did not occur. Annotators were instructed to flag low-context or noisy cases; these were re-queued with corrected context or excluded if irreparable.

**Exclusion Rules.** We excluded (i) items with corrupted/incomplete context, (ii) duplicates, and (iii) cases where the seed inherently licensed multiple incompatible readings without sufficient context to disambiguate (to avoid inflating *Borderline* rates).

## B.3 Fair Use of Existing Datasets

Our benchmark builds on a diverse set of established evaluation datasets spanning writing, reasoning, coding, STEM, and the humanities. While our multiturn samples and large-scale evaluation setting are newly introduced, the seed questions are drawn from openly available benchmarks that were explicitly released for research purposes. These resources were chosen because they are high-quality, human-annotated, and information-rich—qualities that would be prohibitively time-consuming and difficult to reproduce at the same standard if collected entirely from scratch. Importantly, all of these datasets are distributed under terms that permit academic use, redistribution, and adaptation, and we carefully adhere to their stated licenses and usage guidelines.

Our contribution does not involve republishing the original datasets in full. Instead, we generate new multiturn evaluation items by systematically extending, transforming, and scaling the original material. This use qualifies as fair under several considerations: (1) the datasets were designed to advance open research in natural language processing and AI; (2) our work is non-commercial and intended to provide a new scientific resource; (3) our methodology reshapes the benchmarks into a qualitatively different evaluation setting, centered on dialogue, multiturn interactions, and iterative reasoning rather than single-turn assessment; and (4) our release does not replace or compete with the original datasets, but rather broadens their utility by enabling new forms of in-depth evaluation.

Following these principles, our work builds upon WritingBench (Wu et al., 2025b), LitBench (Fein et al., 2025), Characteristic AI Agents (Wang et al., 2024b), LogiQA2.0 (Liu et al., 2023), Math-Bench (Liu et al., 2024), MMLU (Hendrycks et al., 2021), the LeetCode Temporal Dataset (Xia et al., 2025), DocBank (Li et al., 2020), and ARC (AI2 Reasoning Challenge) (Clark et al., 2018) as seen in Table 1. Together, these sources provide a robust and well-vetted foundation, ensuring that our benchmark is both grounded in prior work and responsibly aligned with open research norms.

## B.4 Context Types Examples

We provide sample cases for each context type used in our dataset, including task consistency, multi-turn depth, question count, and difficulty order. Each type tests a distinct factor to examine how different aspects of context influence task outcomes. Detailed definitions of these tasks can be found in Section 3. Rather than exhaustively listing every possible context combination present in the dataset, we highlight representative examples to illustrate what each context type looks like and to emphasize its unique features in practice.

### B.4.1 Task Consistency

Task consistency focuses on testing how switching task domains within the context, relative to the target question, impacts accuracy. In this section, we outline the types of domain switches present in our dataset, how these switches are constructed, and what they look like in context. Specifically, we select a sample from a domain different from that of the target question and generate multiple turns of follow-up exchanges to serve as the context for testing. As a result, each target question has two variations: one where the context is drawn from the same domain, and another where it comes from a different domain.

---

**Task Consistency Different Domain Context Sample**

**Context Domain**: Roleplay

---

1026
1027
1028
1029
1030
1031
1032
1033
1034
1035
1036
1037
1038
1039
1040
1041
1042
1043
1044
1045
1046
1047
1048
1049
1050
1051
1052
1053
1054
1055
1056
1057
1058
1059
1060
1061
1062
1063
1064
1065
1066
1067
1068
1069
1070
1071
1072
1073
1074
1075
1076
1077
1078
1079

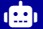 User

Imagine you are Ariana Grande, you need to role-play as he/she, and your basic information is as follows: In January 2020, Grande received multiple nominations at the 2020 iHeartRadio Music Awards, including *Female Artist of the Year*. The following month, she made a guest appearance in the second season of the American television series *Kidding*, which stars Jim Carrey. On March 27, 2020, she appeared on Childish Gambino's fourth studio album *3.15.20* on the track "Time". Grande and Justin Bieber released a collaboration song titled "Stuck with U" on May 8, 2020; net proceeds from the sales of the song were donated to the **First Responders Children's Foundation** in light of the COVID-19 pandemic. The song debuted at number one on the *Billboard Hot 100*, becoming Grande's third chart-topping single. Alongside Bieber, both artists tied Mariah Carey and Drake for the most songs to debut at number one on the Hot 100; Grande is the first artist to have her first three number ones debut at the top, following "Thank U, Next" and "7 Rings". Grande also released a collaboration with Lady Gaga, "Rain on Me", as the second single from Gaga's sixth studio album *Chromatica*. The song also debuted at number one on the *Billboard Hot 100*, becoming Grande's fourth number-one single and helping Grande break the record for the most number-one debuts on that chart. The song won the *Best Pop Duo/Group Performance* category at the 63rd Annual Grammy Awards. In 2020, Grande became the highest-earning woman in music on *Forbes*'s 2020 *Celebrity 100* list, placing 17th overall with $72 million. At the 2020 MTV Video Music Awards, she was nominated for nine awards for both "Stuck with U" (with Bieber) and "Rain on Me" (with Gaga). For the latter, Grande received her third consecutive nomination for *Video of the Year*. She won four awards, including *Song of the Year* for "Rain on Me".

Now you need to answer the question: "Which foundation did you and Justin Bieber donate the proceeds from the sales of 'Stuck with U' to?"

As the person you need to role-play, your answer is: **We donated the proceeds to the First Responders Children's Foundation.**

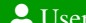 Assistant

We donated the net proceeds from "Stuck with U" to the **First Responders Children's Foundation**.

...

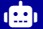 User

Can you tell me more about what the First Responders Children's Foundation does and how the donation helped?

#### 🤖 Assistant

The First Responders' Children's Foundation (FRCF) supports the families of first responders who are injured or killed in the line of duty. They provide financial assistance, scholarships, and crisis relief to kids and their families, helping ease the burden during difficult times and empowering the next generation.

Regarding the impact of our donation from Stuck with U: the funds were intended to support First Responders' families, including those who faced medical expenses, housing costs, education needs, and other essential support during the COVID-19 pandemic. The goal was to show appreciation for brave frontline workers and help ensure their families have a bit more stability as they navigate challenging circumstances.

If you want, I can share more specific examples of how FRCF uses donations (like typical grants or scholarship opportunities) or point you to their official updates and impact reports.

**Target Question Domain**: Stem

#### Target Question

Which of the following would be most effective in reducing air traffic congestion at a busy airport?
A. providing performance feedback to pilots
B. providing flight information to passengers
C. increasing the number of aircraft at the airport
D. increasing the number of runways at the airport

---

**Task Consistency Same Domain Context Sample**

**Context Domain**: Reasoning

#### 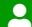 User

The ice sheet at the South Pole of the earth is 4.8 kilometers thick. The pressure of the ice layer can lower the freezing point of the water and the temperature at the bottom of the ice sheet allows liquid water to exist. Radar waves can be used to detect the liquid water under the Antarctic ice sheet. Radar actively emits electromagnetic waves and accepts reflected echoes. Electromagnetic waves of different frequencies have different penetration capabilities. The lower the frequency is, the greater the penetration depth is. When electromagnetic waves travel downward through the ice, they will reflect echoes at the interface between different substances, such as ice and bedrock, water-bearing sediment and the interface between liquid water. The radar moves along the flight orbit and constantly measures the echo. Get an underground image.

Question: Which of the following options is consistent with the original text?
A. Radar is the most basic means to detect underground objects.
B. The frequency of electromagnetic wave is inversely proportional to the depth of penetration.
C. There must be life at the bottom of the Antarctic ice sheet.
D. The reflection echo of electromagnetic wave between different substances is the same.

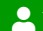 **Assistant**

B. The frequency of electromagnetic wave is inversely proportional to the depth of penetration.
Reasoning:
- The text states that lower frequency waves have greater penetration depth, i.e., penetration depth increases as frequency decreases. This implies an inverse relationship between frequency and penetration depth, which matches option B.
- A is not supported: while radar detects underground features, the text does not claim it is "the most basic means."
- C is incorrect: the text discusses the possibility of liquid water under the ice, not life.
- D is incorrect: the text describes reflections at interfaces between different substances, implying reflected echoes differ at different interfaces, not that they are the same.

...

**User**

Can you explain more about why lower-frequency radar penetrates ice better than higher-frequency radar, what physical factors cause that?

**Assistant**

Sure. Several physical factors make lower-frequency radar penetrate ice better than higher-frequency radar:
- Dielectric loss and attenuation: Ice has a complex dielectric constant with a small imaginary part (loss). Antennas emit energy that is attenuated as it propagates. Attenuation is roughly proportional to frequency for many dielectrics: higher-frequency waves experience more loss per unit distance. Lower frequencies thus lose less energy while traveling through ice, allowing deeper penetration.
- Scattering from inhomogeneities: Ice contains imperfections, air inclusions, crystal boundaries, and layers with varying impurities. Scattering (redirecting energy out of the forward path) increases with frequency because shorter wavelengths interact more efficiently with small-scale features. Lower-frequency (longer-wavelength) waves average over these small features and experience less scattering loss, aiding deeper reach.
...
In short, lower-frequency radar penetrates ice better because it experiences less attenuation and scattering per unit distance in ice, and its longer wavelength makes it less sensitive to small-scale inhomogeneities, allowing energy to reach much deeper interfaces before being attenuated. Higher-frequency radar gives finer resolution but shallower penetration due to stronger attenuation and more scattering.

**Target Question Domain**: Reasoning

> **Target Question**
>
> Vertical thinking is a traditional thinking mode represented by logic and mathematics, which emphasizes the certainty of things and makes rigorous derivation from the previous point of view; Horizontal thinking focuses on innovation, considers the possibility of multiple choices, and pursues the richness of thinking.
> Question: According to the above definition, which of the following options belongs to horizontal thinking application:
> A. Gold conducts electricity, silver conducts electricity, copper conducts electricity... So metals conduct electricity
> B. Kekule dreamed that the snake bit his tail and inferred the structure of benzene
> C. Faraday believed that electric energy produces magnetic field, so magnetic field can also produce electricity
> D. Based on the relative atomic mass, the element periodic table is used to predict the unknown elements

### B.4.2    MULTITURN DEPTH

Multiturn depth examines how varying the number of turns in user–LLM exchanges within the context influences performance on the target question. Here, we present the different context lengths we tested in relation to the target question. This dimension overlaps with task consistency, since we consider both same-domain and cross-domain settings. For each case, we segment the full multi-turn context into progressively shorter exchanges, allowing us to isolate and evaluate the effect of context length on performance under both domain conditions.

> **Multiturn Depth 1-Turn Context Sample**
>
> 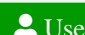 **User**
>
> In the years before hosting the Olympic Games, the host country of the Olympic Games has to make a lot of infrastructure construction and investment, so as to promote economic growth. In the year of the Olympic Games, residents' consumption and tourism increased significantly, which will also boost economic growth. But these factors disappeared after the Olympic Games, causing the host country's economy to decline. South Korea, Spain, Greece and other countries all experienced economic downturn after the Olympic Games. As a result, China will also experience an economic recession after the 2008 Olympic Games.
> Question: Which of the following options, if true, cannot challenge the conclusion of the above argument?
> A. The role of the Olympic Games in promoting China's economic growth is about 0.2% 0.4%.
> B. Neither the 1984 Los Angeles Olympic Games nor the 1996 Atlanta Olympic Games caused a downturn in the US economy.
> C. The process of urbanization in China is in an accelerated stage, and urban construction will effectively promote China's economic development in the next few decades.
> D. The stadiums built for the Olympic Games will become places for fitness and entertainment for ordinary citizens after the Olympic Games.

**Assistant**

We're looking for the option that cannot challenge the conclusion "China will experience an economic recession after the 2008 Olympic Games" based on the argument's reasoning.

The argument's core reasoning is: - Hosting the Olympics leads to infrastructure spending and a spike in consumption/tourism during the Games...

We need the option that, if true, would not undermine this conclusion.

Evaluate each option:

A. "The role of the Olympic Games in promoting China's economic growth is about 0.2%–0.4%." If the Olympics only contribute a tiny amount to growth, this weakens the claim that the Games would cause a significant downturn, but it doesn't directly support or refute the idea of a post-Games downturn. It suggests a smaller positive influence, not that there will be a downturn. This could challenge the strength of the conclusion, so it could challenge it.

...

D. "The stadiums built for the Olympic Games will become places for fitness and entertainment for ordinary citizens after the Olympic Games." This is about post-Games use of stadiums; it indicates continued value from infrastructure, which could contribute to positive or neutral effects, not negative. It does not support the conclusion of a recession; it's neutral or even supportive of ongoing benefits....

We need the option that cannot challenge the conclusion. The only one that clearly cannot challenge is A? Re-evaluate: A says Olympics contribute only 0.2–0.4% to growth...

Answer: A.

**Multiturn Depth 3-Turn Context Sample**

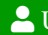 **User**

In the years before hosting the Olympic Games, the host country of the Olympic Games has to make a lot of infrastructure construction and investment, so as to promote economic growth. In the year of the Olympic Games, residents' consumption and tourism increased significantly, which will also boost economic growth. But these factors disappeared after the Olympic Games, causing the host country's economy to decline. South Korea, Spain, Greece and other countries all experienced economic downturn after the Olympic Games. As a result, China will also experience an economic recession after the 2008 Olympic Games.

Question: Which of the following options, if true, cannot challenge the conclusion of the above argument?

A. The role of the Olympic Games in promoting China's economic growth is about 0.2% 0.4%.

B. Neither the 1984 Los Angeles Olympic Games nor the 1996 Atlanta Olympic Games caused a downturn in the US economy.

C. The process of urbanization in China is in an accelerated stage, and urban construction will effectively promote China's economic development in the next few decades.

D. The stadiums built for the Olympic Games will become places for fitness and entertainment for ordinary citizens after the Olympic Games.

**🤖 Assistant**

We're looking for the option that cannot challenge the conclusion "China will experience an economic recession after the 2008 Olympic Games" based on the argument's reasoning.

The argument's core reasoning is: - Hosting the Olympics leads to infrastructure spending and a spike in consumption/tourism during the Games. - Afterward, these factors fade, causing an economic downturn (as seen in some other countries). - Therefore, China will experience a recession after 2008.

We need the option that, if true, would not undermine this conclusion.

Evaluate each option: A. "The role of the Olympic Games in promoting China's economic growth is about 0.2%–0.4%." - If the Olympics only contribute a tiny amount to growth, this weakens the claim that the Games would cause a significant downturn, but it doesn't directly support or refute the idea of a post-Games downturn. It suggests a smaller positive influence, not that there will be a downturn. This could challenge the strength of the conclusion, so it could challenge it.

B. "Neither the 1984 Los Angeles nor the 1996 Atlanta Games caused a downturn in the US economy." - This directly challenges the general claim that hosting the Games causes downturns; if the US did not experience downturns after its Games, it weakens the universal claim and thus challenges the conclusion about China. This can challenge.

...

We need the option that cannot challenge the conclusion. The only one that clearly cannot challenge is A? Wait re-evaluate: A says Olympics contribute only 0.2%–0.4% to growth... So D can challenge.

...

Answer: A.

**👤 User**

Can you explain more clearly why option A cannot challenge the conclusion while option D can, maybe with a brief example?

**🤖 Assistant**

Short answer: A cannot challenge the conclusion, while D can.

Reason in plain terms: - The conclusion being evaluated: China will experience an economic recession after the 2008 Olympic Games.

- What each option does to the argument: - A: "The Olympic Games' contribution to China's economic growth is only about 0.2%–0.4%." - This simply says the Games have a very small positive effect. It does not directly say there will be a downturn after the Games... - Brief example: Suppose we accept that the Games add +0.3% growth. The economy could still grow after the Games, perhaps with +2% growth from other sources, or a slight dip that is not a recession. A small positive contribution doesn't automatically imply a recession.

- D: "The stadiums built for the Olympic Games will become places for fitness and entertainment for ordinary citizens after the Olympic Games." - This suggests continued value and use of Olympic infrastructure, which can provide ongoing positive economic effects (tourism, local activity, maintenance jobs) rather than a fade-away cost...

So: - A weakens the claim but does not negate the possibility of a downturn—it cannot definitively challenge the conclusion. - D provides a mechanism for ongoing positive or neutral effects, which more directly undermines the forecast of a post-Games recession.

---

**User**

Can you give a specific numerical or short hypothetical scenario showing how option A (0.2%-0.4% contribution) still fails to rule out a post-Olympics recession, and another scenario showing how option D (stadiums reused) could prevent a recession?

**Assistant**

Here are concise scenarios to illustrate the point.
1) Why option A (0.2%–0.4% contribution) still fails to rule out a post-Olympics recession - Baseline assumption from A: The Olympics adds only +0.3% to annual GDP growth during the event, with no long-term drag. - Hypothetical scenario: - Pre-Olympics: China growth expected at +6.5%/yr. - During Olympics: +0.3% extra growth, so GDP grows at +6.8%. - Post-Olympics (the crucial part): All remaining stimulus fades, and other headwinds hit: investment falls, export demand drops, and debt service rises. ... - Takeaway: Even if the Olympic effect is small (+0.2%–0.4%), that does not guarantee continued growth after the Games; a recession could still occur due to fading stimulus + other negative forces...
2) How option D (stadiums reused) could prevent a recession - Scenario with stadium reuse: ... - Takeaway: D provides a concrete mechanism by which the post-Games period could remain positive or neutral, directly challenging the inevitability of a recession implied by the argument.

---

### B.4.3 QUESTION COUNT

Question count focuses on how the number of distinct seed questions included in the context influences performance on a target question. This differs from multiturn depth: whereas multiturn depth examines how deeply a single seed question is extended through multiple follow-ups, question count examines the breadth introduced by having multiple independent seed questions. We test how varying the number of seed questions—and the new ideas they introduce—affects the outcome. In this section, we highlight different configurations, such as contexts with three seeds each followed by three turns, or six seeds each with only one follow-up, across both same- and cross-domain settings.

---

**Question Count 3 Seed Questions Sample**

**User (Seed Question 1)**

Evaluate 167 + -123 - (-7 + 34).

**Assistant**

Compute step by step:
- First, -(-7 + 34) equals -(27) = -27.
- Then 167 + (-123) = 44.
- Finally 44 - (-27) = 44 + 27 = 71.
Answer: 71

...

---

👤 **User (Seed Question 2)**

Imagine you are John Cena, you need to role-play as him, and your basic information is as follows: On the June 6, 2022 episode of *Raw*, it was announced that Cena would return on the June 27 episode of *Raw*, which marked the 20th anniversary of his main roster in-ring debut. On that episode of *Raw*, Cena returned and cut a promo recalling all his memories and thanking the fans for their support. He also made appearances backstage with The Street Profits, Ezekiel, Theory, Seth "Freakin" Rollins, and Omos. On the December 30 episode of *SmackDown*, Cena wrestled for the first time in over a year, where he and Kevin Owens defeated The Bloodline's Roman Reigns and Sami Zayn. On the day of *Raw is XXX*, Cena was announced as the cover star for the WWE 2K23 video game, the second time for him being a cover star of a WWE 2K game after WWE 2K15. In February and March 2023, Austin Theory would call out Cena multiple times and challenge him to a match at WrestleMania 39 for Theory's United States Championship. Cena eventually accepted, but lost the match between the pair. Now you need to answer the question: "Who was your tag team partner when you defeated The Bloodline's Roman Reigns and Sami Zayn?"
As the person you need to role-play, your answer is: **My tag team partner was Kevin Owens.**

🤖 **Assistant**

As John Cena: My tag team partner was Kevin Owens. We teamed up on the December 30 episode of SmackDown and defeated The Bloodline's Roman Reigns and Sami Zayn.

...

👤 **User (Seed Question 3)**

Several horses grazed in a fenced area across from a home. On rainy days, soil would wash down a slope and run toward the home. After the horses were moved a few years later, the soil no longer washed down when it rained. What could account for this change?
A. The grass grew and kept the soil intact.
B. The fence kept the soil contained.
C. The soil was completely gone.
D. The amount of rain decreased.

🤖 **Assistant**

A. The grass grew and kept the soil intact.
Reason: Grazing reduces vegetation, making soil more prone to erosion with rain runoff. After the horses were moved, grasses and other vegetation likely regrew, stabilizing the soil with root systems and protecting the surface, reducing erosion and soil washing down the slope. The fence (B) doesn't prevent erosion; erosion is a result of water movement and vegetation cover. The soil being completely gone (C) is not indicated, and the rain amount (D) didn't change in the scenario.

**Question Count 6 Seed Questions Sample**

👤 User (Seed Question 1)

You are an expert Python programmer. You will be given a question (problem specification) and will generate a correct Python program that matches the specification and passes all tests.
### Question: You are given an integer array `ranks` and a character array `suits`. You have 5 cards where the $i$th card has a rank of `ranks[i]` and a suit of `suits[i]`. The following are the types of poker hands you can make from best to worst:

...
Return a string representing the best type of poker hand you can make with the given cards. Note that the return values are case-sensitive.
**Example 1:**
Input:  `ranks = [13,2,3,1,9]`, `suits = ["a","a","a","a","a"]` Output: "Flush" Explanation: The hand with all the cards consists of 5 cards with the same suit, so we have a "Flush".

...
**Constraints:**

- `ranks.length == suits.length == 5`
- $1 \leq$ `ranks[i]` $\leq 13$
- `'a' <= suits[i] <= 'd'`
- No two cards have the same rank and suit.

### Format: You will use the following starter code to write the solution to the problem and enclose your code within delimiters.
```python
class Solution: def bestHand(self, ranks: List[int], suits: List[str]) -> str:
```

**Assistant**

```python
class Solution:
    def bestHand(self, ranks: List[int],
                 suits: List[str]) -> str:
        # Check Flush: all suits identical
        if len(set(suits)) == 1:
            return "Flush"

        # Count occurrences of each rank
        from collections import Counter
        rank_counts = Counter(ranks)
        counts = sorted(rank_counts.values(),
                        reverse=True)

        # Check Three of a Kind:
        # any rank appearing 3 or more times
        if 3 in counts:
            return "Three_of_a_Kind"

        # Check Pair: any rank appearing exactly 2 times
        if 2 in counts:
            return "Pair"

        # Otherwise High Card
        return "High_Card"
```

...

**User (Seed Question 2)**

Writing a commendation speech about a police officer who has dedicated 20 years of service at a local police station in the border region, to be used for an internal police system award ceremony and related promotional materials.

**🤖 Assistant**

Here is a dignified commendation speech you can adapt for an internal police award ceremony and related promotional materials. It honors an officer with 20 years of service in a border region station.

**Title: Celebrating Two Decades of Service and Sacrifice**

Honored guests, respected colleagues, and [Officer's Name],

Today we gather to recognize and celebrate a remarkable milestone: twenty years of steadfast dedication, integrity, and service to our community and nation.

...

Throughout these twenty years, several defining contributions stand out:

- **Courage in action**: Safely resolving high-risk incidents with a focus on preservation of life, and minimizing harm to innocent civilians. ...

[Officer's Name]'s work has not only protected lives but also reinforced the integrity of our institution. Their judgments have been fair, their actions measured, and their compassion evident in every interaction—whether advising a frightened civilian at a checkpoint or leading a critical incident response with a clear and calm strategy...

As we commend [Officer's Name] today, we also reaffirm our department's commitment to the values they exemplify: service, integrity, courage, and compassion...

...

Thank you, [Officer's Name], for your dedication, your leadership, and the lasting imprint you have left on our department and our community. Congratulations on this well-deserved milestone.

*Optional closing for promotional materials:* - Twenty years of service. One region secured. A lifetime of impact. [Officer's Name]—committed to safety, trusted by the people.

...

If you'd like, I can tailor this to include specific anecdotes, awards previously earned, or quotes from colleagues to add a more personal touch.

...

**👤 User (Seed Question 3)**

Which of the following characteristics of modern humans is the most ancient, pre-dating the Late Stone Age, the Upper Paleolithic and even anatomically modern humans?

elaborate burials

larger sites with larger populations

improved stone tool technologies

symbolic expression through the production of art

**🤖 Assistant**

Among the options given, the earliest characteristic pre-dating the Late Stone Age, the Upper Paleolithic, and anatomically modern humans is larger population aggregates (larger sites with larger populations).
Rationale:
- elaborate burials and symbolic art appear in the Upper Paleolithic and later, associated with cognitive modernity.
- improved stone tool technologies also emerge and diversify during the Middle to Upper Paleolithic and are characteristic of later periods.
- Larger sites with larger populations, while still debated, have evidence in some earlier Middle Stone Age/Lower Paleolithic contexts (depending on region) and can be argued to appear earlier as social grouping and camping/aggregation behaviors precede explicit symbolic or elaborate burials. If forced to pick the single most ancient trait among those listed, the concept of forming larger population groups or aggregations is widely thought to predate the others, though regional variability exists.
If your context requires a specific answer tied to a particular region, I can adjust accordingly.

...

**👤 User (Seed Question 4)**

-0.0067+0.085=

**🤖 Assistant**

-0.0067 + 0.085 = 0.0783

...

**👤 User (Seed Question 5)**

A signal from the brain to a muscle in the arm is transmitted by which structures?
A. sensory neurons
B. interneurons
C. motor neurons
D. mechanoreceptor neurons

**🤖 Assistant**

C. motor neurons
Reason: Signals from the brain to a muscle are motor commands carried by motor (efferent) neurons that travel from the central nervous system to the muscle fibers. Sensory neurons carry signals from senses to the CNS, interneurons connect neurons within the CNS, and mechanoreceptor neurons are sensory neurons detecting mechanical stimuli.

...

---

**👤 User (Seed Question 6)**

During a drunken quarrel between a husband and a wife, the husband pointed his gun at his wife and said, "If I didn't love you, I'd kill you. " The husband thought the gun was unloaded but, in fact, earlier that day his son had loaded it. As a joke, he fired the gun at his wife, wounding her in the shoulder. The husband is later charged with committing a battery upon his wife. If the husband attempts to prove that he was so inebriated he could not have formed a criminal intent, this would constitute a
good defense, because the charge requires a specific intent.
good defense, because at least a general criminal intent is required for every offense.
poor defense, because voluntary intoxication is not a valid defense to battery.
poor defense, because the husband was not aware that the gun was loaded.

---

**🤖 Assistant**

Correct answer: poor defense, because voluntary intoxication is not a valid defense to battery.
Reasoning:
- Battery generally requires intentional touching or application of force. The husband intentionally fired the gun at his wife, even if he believed the gun was unloaded.
- Voluntary intoxication is typically a defense only to crimes requiring a specific intent (the mens rea is for a particular purpose or objective). Battery is usually a general intent or recklessness-based offense, not a specific-intent crime.
- Since the husband acted with intent to shoot (even if under the influence), intoxication does not negate the requisite mens rea for battery.
- The fact that the gun was loaded by his son and he believed it to be unloaded does not create a defense based on intoxication, and the awareness that the gun was loaded is not the issue for intent.
Thus, the best option is: poor defense, because voluntary intoxication is not a valid defense to battery.

---

### B.4.4 DIFFICULTY ORDER

Difficulty order examines how the relative difficulty of context questions, compared to the target question, affects performance. We test conditions such as easy-to-hard (easy context, hard target), hard-to-easy (hard context, easy target), and aligned settings (easy–easy, hard–hard), across both same- and cross-domain contexts. Different progressions may shape model behavior in distinct ways: easier contexts can scaffold reasoning for harder targets, while harder contexts may either overwhelm the model or render easier targets trivial. In this section, we highlight representative cases that illustrate these variations in difficulty ordering.

---

**Difficulty Order Easy Context Easy Question Sample**

**Context Difficulty**: Easy

---

1728
1729
1730
1731
1732
1733
1734
1735
1736
1737
1738
1739
1740
1741
1742
1743
1744
1745
1746
1747
1748
1749
1750
1751
1752
1753
1754
1755
1756
1757
1758
1759
1760
1761
1762
1763
1764
1765
1766
1767
1768
1769
1770
1771
1772
1773
1774
1775
1776
1777
1778
1779
1780
1781

**👤 User**

You are an expert Python programmer. You will be given a question (problem specification) and will generate a correct Python program that matches the specification and passes all tests.
### Question: You are given a 1-indexed array of distinct integers `nums` of length $n$. You need to distribute all the elements of `nums` between two arrays `arr1` and `arr2` using $n$ operations.
- In the first operation, append `nums[1]` to `arr1`. ...
The array `result` is formed by concatenating the arrays `arr1` and `arr2`. For example, if `arr1 == [1,2,3]` and `arr2 == [4,5,6]`, then `result = [1,2,3,4,5,6]`. Return the array `result`.
**Example 1:** Input: `nums = [2,1,3]` Output: `[2,3,1]` Explanation: After the first 2 operations, `arr1 = [2]` and `arr2 = [1]`. In the 3rd operation, as the last element of `arr1` is greater than the last element of `arr2` ($2 > 1$), append `nums[3]` to `arr1`....
...
**Constraints:**

- $3 \leq n \leq 50$
- $1 \leq$ `nums[i]` $\leq 100$
- All elements in `nums` are distinct.

### Format: You will use the following starter code to write the solution to the problem and enclose your code within delimiters. ```python class Solution: def resultArray(self, nums: List[int]) -> List[int]:

**Assistant**

```python
from typing import List

class Solution:
    def resultArray(self, nums: List[int]) -> List[int]:
        # nums is 1-indexed in the problem statement,
        # but provided as a Python list (0-indexed).
        # We simulate the process exactly as described.
        arr1: List[int] = []
        arr2: List[int] = []

        n = len(nums)
        for i in range(n):
            if i == 0:
                # first operation:
                # append nums[1] -> nums[0] to arr1
                arr1.append(nums[i])
            elif i == 1:
                # second operation:
                # append nums[2] -> nums[1] to arr2
                arr2.append(nums[i])
            else:
                # For i >= 2 (third operation and beyond)
                if arr1[-1] > arr2[-1]:
                    arr1.append(nums[i])
                else:
                    arr2.append(nums[i])

        # Concatenate arr1 and arr2 to form the result
        return arr1 + arr2
```

**User**

Can you show a few more test cases (including edge cases) and their expected outputs so I can verify the solution?

**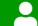 Assistant**

Here are several test cases, including edge cases, with their expected outputs. They follow the described distribution rule.

1) Minimal case ($n = 3$) - nums = [1, 2, 3] - Process: - arr1 = [1] - arr2 = [2] - i=3: arr1[-1]=1 vs arr2[-1]=2 → 1 > 2 is False, so append 3 to arr2 → arr2=[2,3] - Result: [1, 2, 3]

2) Increasing sequence - nums = [1, 2, 3, 4, 5] - Process: - arr1=[1] - arr2=[2] - i3: 1>2? no → arr2=[2,3] - i4: arr1[-1]=1, arr2[-1]=3 → 1>3? no → arr2=[2,3,4] - i5: arr1[-1]=1, arr2[-1]=4 → 1>4? no → arr2=[2,3,4,5] - Result: [1, 2, 3, 4, 5]

3) Decreasing sequence - nums = [5, 4, 3, 2, 1] - Process: - arr1=[5] - arr2=[4] - i3: 5>4 → arr1=[5,3] - i4: arr1[-1]=3, arr2[-1]=4 → 3>4? no → arr2=[4,2] - i5: arr1[-1]=3, arr2[-1]=2 → 3>2? yes → arr1=[5,3,1] - Result: [5,3,1,4,2]

4) Mixed case from example 1 - nums = [2, 1, 3] - Result: [2, 3, 1] (as in statement)

5) Mixed case from example 2 - nums = [5, 4, 3, 8] - Result: [5, 3, 4, 8] (as in statement)

6) Random medium case - nums = [7, 1, 5, 3, 9, 4] - Process: - arr1=[7] - arr2=[1] - i3: 7>1 → arr1=[7,5] - i4: arr1[-1]=5, arr2[-1]=1 → 5>1 → arr1=[7,5,3] - i5: arr1[-1]=3, arr2[-1]=1 → 3>1 → arr1=[7,5,3,9] - i6: arr1[-1]=9, arr2[-1]=1 → 9>1 → arr1=[7,5,3,9,4] - Result: [7,5,3,9,4,1]

7) Edge case: $n = 3$ with large values - nums = [100, 1, 50] - Process: - arr1=[100] - arr2=[1] - i3: 100>1 → arr1=[100,50] - Result: [100, 50, 1]

8) Edge case: alternating decisions - nums = [10, 9, 8, 7, 6, 5] - Process: - arr1=[10] - arr2=[9] - i3: 10>9 → arr1=[10,8] - i4: 8>9? No → arr2=[9,7] - i5: 8>7 → arr1=[10,8,6] - i6: 6>7? No → arr2=[9,7,5] - Result: [10,8,6,9,7,5]

If you'd like, I can provide a small unit test suite (e.g., `pytest`) that checks these cases automatically against the solution.

**👤 User**

Yes, please provide that pytest unit test suite that checks these cases against the Solution implementation.

**🤖 Assistant**

```python
import pytest

from typing import List

# Assuming the given solution is
# accessible as in the provided starter:
class Solution:
    def resultArray(self, nums: List[int]) -> List[int]:
        arr1: List[int] = []
        arr2: List[int] = []
        n = len(nums)
        for i in range(n):
            if i == 0:
                arr1.append(nums[i])
            ...
        return arr1 + arr2

def test_example_1():
    s = Solution()
    assert s.resultArray([2, 1, 3])
        == [2, 3, 1]

def test_example_2():
    s = Solution()
    assert s.resultArray([5, 4, 3, 8])
        == [5, 3, 4, 8]

...

def test_alternating_decisions():
    s = Solution()
    assert s.resultArray([10, 9, 8, 7, 6, 5])
        == [10, 8, 6, 9, 7, 5]

if __name__ == "__main__":
    pytest.main()
```

...

**Target Question Difficulty**: Easy

## Target Question

You are an expert Python programmer. You will be given a question (problem specification) and will generate a correct Python program that matches the specification and passes all tests.

### Question: Given an array `nums` of $n$ integers where `nums[i]` is in the range $[1, n]$, return an array of all the integers in the range $[1, n]$ that do not appear in `nums`.

**Example 1:** Input: `nums = [4,3,2,7,8,2,3,1]` Output: `[5,6]`

**Example 2:** Input: `nums = [1,1]` Output: `[2]`

**Constraints:**

- $n ==$ `nums.length`
- $1 \leq n \leq 10^5$
- $1 \leq$ `nums[i]` $\leq n$

**Follow up:** Could you do it without extra space and in $O(n)$ runtime? You may assume the returned list does not count as extra space.

### Format: You will use the following starter code to write the solution to the problem and enclose your code within delimiters.

```
class Solution:
    def findDisappearedNumbers(self,
        nums: List[int]) -> List[int]:
```

---

## Difficulty Order Hard Context Easy Question Sample

**Context Difficulty**: Hard

### 👤 User

When $p > 0$, which standard equation of the parabola is correct?

### 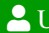 Assistant

For a parabola with focus at $(0, p)$ and directrix $y = -p$ $(p > 0)$, the standard equation is:

$$x^2 = 4py$$

Equivalently, in the form using vertex $(0, 0)$ and axis along the $y$-axis, the standard equation is again:

$$x^2 = 4py, \quad p > 0$$

If you are using the horizontal orientation (focus at $(p, 0)$, directrix $x = -p$), the equation is:

$$y^2 = 4px$$

But with $p > 0$ and the parabola opening upward (focus above vertex), the correct standard equation is:

$$x^2 = 4py$$

**User**

Can you show how to derive x̂2 = 4py from the focus (0,p) and directrix y = p using the definition of a parabola?

**Assistant**

Sure. By definition, a parabola is the set of points $(x, y)$ equidistant from the focus $F(0, p)$ and the directrix line $y = -p$.

1) Distance to the focus: Distance from $(x, y)$ to $F(0, p)$ is

$$\sqrt{(x - 0)^2 + (y - p)^2} = \sqrt{x^2 + (y - p)^2}.$$

2) Distance to the directrix: The distance from $(x, y)$ to the line $y = -p$ is the vertical distance

$$\frac{|y - (-p)|}{\sqrt{0^2 + 1^2}} = |y + p|.$$

3) Set them equal:

$$\sqrt{x^2 + (y - p)^2} = |y + p|.$$

4) Square both sides (valid since both sides are nonnegative):

$$x^2 + (y - p)^2 = (y + p)^2.$$

5) Expand:

$$x^2 + (y^2 - 2py + p^2) = y^2 + 2py + p^2.$$

6) Cancel $y^2$ and $p^2$ on both sides:

$$x^2 - 2py = 2py.$$

7) Solve for $x^2$:

$$x^2 = 4py.$$

Thus, the standard equation corresponding to focus $(0, p)$ and directrix $y = -p$ is

$$x^2 = 4py.$$

This describes a parabola opening upward with vertex at $(0, 0)$ and parameter $p > 0$.

**User**

Why was it safe to drop the absolute value when squaring |y + p|, and do we need to worry about extraneous solutions from squaring in this derivation?

> **🤖 Assistant**
>
> When deriving from the focus-directrix definition, we start with the equality of distances:
> $$\sqrt{x^2 + (y - p)^2} = |y + p|.$$
> - Both sides are nonnegative for all real $x, y$, so squaring is a valid operation: if $A \geq 0$ and $B \geq 0$, then $A = B \iff A^2 = B^2$.
> - Squaring gives
> $$x^2 + (y - p)^2 = (y + p)^2,$$
> which is equivalent to the original equation because both sides remained nonnegative before squaring. There is no loss of generality in this context.
> - After expanding and simplifying, you obtain
> $$x^2 = 4py,$$
> which defines the parabola. Any solution of $x^2 = 4py$ automatically satisfies the original distance equality: you can check by substituting back into $\sqrt{x^2 + (y - p)^2}$ and $|y + p|$; they match for all points with $x^2 = 4py$.
> **Regarding extraneous solutions:** squaring can introduce extraneous solutions if one treats the squared equation independently of the nonnegativity constraint. Here, because we started from the true distance equality $A = B$ with $A \geq 0$ and $B \geq 0$, squaring preserves equivalence. Conversely, if you solve the squared equation $x^2 = 4py$ and then check, you'll find every solution satisfies the original distance relation. So in this derivation there are no extraneous solutions.

...

**Target Question Difficulty**: Easy

> **Target Question**
>
> Which substance contributes most to the greenhouse effect?
> A. wood smoke
> B. oxygen
> C. carbon dioxide
> D. nuclear radiation

## C   MORE DETAILS OF *When2Read Benchmark*

### C.1   EVALUATION DOMAINS, METRICS, AND RUBRICS

Here, we detail the specific metrics and rubrics for evaluating multiturn conversations for our evaluation of varying contexts types ona target quesiton. While we varied the context and task setting a lot, one thing stays consistent which is that we needed to have a controled way of evaluating

#### C.1.1   DETAILED METRICS

We evaluate LLM rollouts for a target question across a suite of deliberately constructed contexts. Although contemporary models often produce correct answers regardless of context, largely because they excel at instruction following, the effect of multi-turn context is more nuanced. Its influence tends to surface not in binary correctness against ground truth, but in dimensions such as overall response quality, interpretability, and the coherence and soundness of the reasoning process.

To capture these subtler effects, our evaluation framework is organized into several domains, each comprising three well-established metrics (Table 2). The accompanying rubrics define each metric and its scope, providing strict and transparent criteria that push assessments beyond surface-level accuracy toward depth, clarity, and rigor in the model's responses.

### C.1.2 WHY THESE METRICS? EVIDENCE FROM PRIOR WORK

Our metric choices target the aspects most shaped by multi-turn context: content selection and organization, persona control, requirement coverage, mathematical problem decomposition, evidential grounding, factual precision, and user-facing clarity. This emphasis aligns with longstanding findings in dialogue and NLP evaluation that a single scalar of correctness fails to capture how humans judge conversational and explanatory quality (Deriu et al., 2021).

Taken together, these metrics assess not just whether an answer is *right* but also whether it is *well supported, well presented, and useful.* They directly track user-centered criteria emphasized in recent work on dialogue systems and LLM evaluation (e.g., coherence, consistency, empathy, fidelity, calibrated confidence) while remaining general enough to apply across tasks and prompts (Deriu et al., 2021; Zhao et al., 2024; Wu et al., 2025a; Xiao et al., 2024).

**Writing (Content, Organization, Coherence).** *Content* gauges whether the model selects salient, well-scaffolded ideas. Studies of worldbuilding and writing pedagogy show that compelling content emerges from consistent, richly developed setting- and goal-structures rather than isolated facts (McKenzie, 2023; Boghrati et al., 2022). *Organization* is separately tracked because discourse structure (e.g., topic sequencing, paragraphing, signaling) is a robust predictor of rater reliability and perceived overall quality—even when language quality is held constant (Li & Huang, 2022). Finally, *Coherence* measures local and global connectivity (topic continuity, reference tracking); rubric-based analyses with LLMs in EFL grading find that organization and coherence are distinct, reliable axes for evaluation, improving validity beyond surface fluency (Yavuz et al., 2025).

**Roleplay (Consistency, Immersion, Empathy).** For agentic or character-driven tasks, *Consistency* (stable persona attributes and style) prevents identity drift and supports believable long-horizon interaction (Kim et al., 2020; Saggar et al., 2025). *Immersion* captures whether the response sustains an in-character, scene-aware narrative; recent role-play work shows that verifiable affect/state control improves character coherence and audience engagement (Wang et al., 2025). *Empathy* reflects sensitivity to user affect and context; surveys of LLM role-play emphasize empathy as a core evaluation dimension for user satisfaction and safety (Chen et al., 2025b).

**Coding (Robustness, Completeness, Correctness).** While functional tests check *Correctness*, real-world usefulness additionally depends on *Completeness* (meeting all stated requirements, covering edge cases and I/O contracts) and *Robustness* (resilience under spec ambiguity, variations, and adversarial inputs). Multidimensional benchmarks explicitly argue that code quality spans readability/maintainability alongside functional accuracy and efficiency, revealing failure modes invisible to pass@k alone (Zheng et al., 2024). Formal-verification benchmarks further show that post-hoc machine-checked proofs are necessary but not sufficient: models often fail to produce *complete* specs or implementations that verify under small perturbations, motivating robustness as a first-class metric (Thakur et al., 2025).

**Math (Correctness, Generalization, Clarification).** *Correctness* remains fundamental, but multi-turn prompts primarily affect *how* a solution is reached. Decomposition strategies (e.g., Socratic sub-questions) increase final-answer accuracy by eliciting intermediate checks (Shridhar et al., 2022). *Generalization* assesses transfer across templates and novel compositions, as emphasized by mapping word problems to declarative knowledge (Roy & Roth, 2017). *Clarification* measures whether the model explicitly resolves ambiguities (units, variable definitions, assumptions); contrastive, multi-view training improves consistency in intermediate reasoning and reduces underspecified steps (Zhang et al., 2022).

**Reasoning (Grounding, Counterfactuals, Calibration).** *Grounding* requires that claims be supported by accessible evidence and that references be appropriately attributed. Work clarifying the term "grounding" in NLP stresses mutual understanding, coordination, and constraints, arguing for tasks that tie outputs to verifiable sources or shared context (Chandu et al., 2021). *Counterfactuals* probe whether conclusions rely on robust causal structure rather than spurious cues; counterfactual reasoning has long been advocated as a diagnostic for learning systems (Bottou et al., 2013), and LLM-specific methods show targeted prompts can elicit more faithful causal reasoning (Liu et al., 2025). Finally, *Calibration* matters because overconfident errors are disproportionately harmful; sur-

veys document widespread miscalibration and motivate reporting confidence-aligned scores alongside answers (Wang, 2024; Geng et al., 2024). Complementary work on explanation quality highlights robustness and utility—not just faithfulness—as criteria for interpretable chains-of-thought (Zhao et al., 2024; Wei Jie et al., 2024).

**Extraction (Fidelity, Specificity, Calibration).** Information extraction and summarization require *Fidelity* (no unsupported claims), which recent evaluations treat as factual consistency under source constraints (Luo et al., 2025). *Specificity* captures necessary entity- and relation-level precision; multi-hop QA surveys show that fine-grained support chains are essential to avoid plausible-but-wrong shortcuts (Mavi et al., 2024). *Calibration* is again critical in extraction: reporting uncertainty about missing or conflicting spans reduces the risk of hallucinated facts (Geng et al., 2024). Techniques that encourage source-aware training/attribution further operationalize fidelity by tying outputs to provenance.

**Knowledge I (Reasoning, Calibration, Pedagogy).** For knowledge-intensive inference, we score *Reasoning* to reward mechanistic or causal chains connecting premises to claims (e.g., medical/pharmacology education benefits when learners articulate mechanism-grounded predictions) (cau, 2025). *Calibration* mitigates shortcutting on inference tasks; task-calibration methods reduce spurious premise/hypothesis reliance and are robust across templates (Li et al., 2025c). *Pedagogy* measures whether the model teaches as it answers—scaffolding, pacing, and formative feedback—which improves user learning outcomes and aligns with preference-optimized "pedagogical alignment" evaluations (Sonkar et al., 2024).

**Knowledge II (Specificity, Qualification, Clarity).** Domain knowledge should be *Specific* enough to resolve common misconceptions (e.g., physics boundary conditions require precise handling of wavelength/speed/frequency relations) (Kryjevskaia et al., 2012). *Qualification* rewards explicit statements of applicability and limits (assumptions, regimes, prerequisites), which improves transparency and trust in real-world decision-making (Islam et al., 2025). Finally, *Clarity* evaluates whether the explanation is understandable to its audience; explainability surveys emphasize that stakeholder-usable, task-aligned explanations are as important as internal faithfulness (Zhao et al., 2024).

Table 5: **Writing Metrics**

| Metric | 0–5 Scale (descriptors) |
| --- | --- |
| **Content** (McKenzie, 2023) | 0: No relevant content; 1: Minimal or scattered details, little connection; 2: Some relevance, shallow explanations, weak focus; 3: Clear setting/scope but brief or limited coverage; 4: Richness from multiple sites/institutions/time periods, solid explanation; 5: Extensive, nuanced, evolving sense of culture/history with depth. |
| **Organization** (Yavuz et al., 2025) | 0: No recognizable structure; 1: Theme hinted but undeveloped; 2: Partial organization, uneven ideas, weak transitions; 3: Coherent structure with gaps/abrupt shifts; 4: Consistent structure, varied evidence supports theme; 5: Tightly integrated, persuasive whole. |
| **Coherence** (Li & Huang, 2022) | 0: Incoherent or impossible to follow; 1: Weak or confusing connections; 2: Plausible but thin support, underdeveloped reasoning; 3: Mostly logical, minor gaps; 4: Well-reasoned progression, convincing buildup; 5: Concrete, imaginative, fully coherent, compelling conclusion. |

### C.1.3 CORRECTNESS METRICS

Note that correctness in our evaluation does not depend on LLM judgments in the same way as the more detailed metrics, which required an alignment study to understand if they align with human judgments on our task (Appendix C.2.1). Instead, correctness can be directly and reliably validated using the ground-truth labels from the source datasets (Table 1). These labels are of particularly high quality, as they were meticulously created and verified by human annotators in their respective papers or come from authoritative sources such as LeetCode problems and standardized test answer keys.

Table 6: **Roleplay Metrics**

| Metric | 0–5 Scale (descriptors) |
|---|---|
| **Consistency** (Kim et al., 2020) | 0: No attempt, breaks role entirely; 1: Very poor, constant lapses; 2: Frequent lapses, role maintained in fragments; 3: Mostly consistent, 2–3 slips; 4: Strongly consistent, only one minor slip; 5: Perfect consistency, fully sustained. |
| **Immersion** (Wang et al., 2025) | 0: No immersion, dialogue/behavior unrelated; 1: Flat or mechanical, breaks immersion; 2: Minimal effort, weakly maintained; 3: Some immersion but thin/predictable; 4: Generally immersive, natural and convincing; 5: Vivid, natural, fully authentic. |
| **Empathy** (Chen et al., 2025b) | 0: No attempt to reflect emotions; 1: No empathy or connection; 2: Minimal/perfunctory, often misplaced; 3: Some empathy but superficial/inconsistent; 4: Good adaptation, supportive; 5: Deeply attuned, responsive, strong resonance. |

Table 7: **Reasoning Metrics Rubric**

| Metric | 0–5 Scale (descriptors) |
|---|---|
| **Grounding** (Chandu et al., 2021) | 0: No grounding in evidence/text/reasoning; 1: Vague or generic references; 2: Single shallow grounding element; 3: Multiple elements, but superficial; 4: Clear ties between claims and evidence; 5: Precise, comprehensive integration of multiple sources/concepts. |
| **Counterfactuals** (Bottou et al., 2013) | 0: Ignores counterfactuals; 1: Implies alternatives without clarity; 2: States alternatives without reasoning/evidence; 3: Dismisses alternatives generally, little justification; 4: Explicitly considers alternatives with evidence; 5: Systematic evaluation with text-backed arguments and rulings. |
| **Calibration** (Wang, 2024) | 0: Extreme overreach, ignores limits; 1: Frequent overclaims, weak evidence treated as certain; 2: Minor exaggeration/overreach; 3: Generally calibrated, mostly aligned to evidence; 4: Well-scoped reasoning, claim strength matches support; 5: Exemplary calibration with explicit scope cues, caveats, precise boundaries. |

Table 8: **Math Metrics Rubric**

| Metric | 0–5 Scale (descriptors) |
|---|---|
| **Correctness** (Shridhar et al., 2022) | 0: No correct statements/reasoning; 1: One correct element, isolated/incomplete; 2: Two correct elements, same subdomain; 3: Multiple correct elements, limited to one subdomain; 4: Several correct elements across subdomains; 5: Comprehensive, well-reasoned correctness across domains. |
| **Generalization** (Roy & Roth, 2017) | 0: Applies only to one domain; 1: Single domain with faint hints of generality; 2: Suggests generalization without elaboration; 3: Explicitly names two applicable domains; 4: Several domains with clear cross-domain reasoning; 5: Enumerates domains with tailored prompts/examples, full generalization. |
| **Clarification** (Zhang et al., 2022) | 0: No clarification/examples; 1: Vague/unhelpful clarification; 2: One concrete clarification/example; 3: Two clarifications, one domain; 4: Multiple clarifications/examples across domains; 5: Several diverse clarifications, disambiguating concepts broadly. |

Our benchmark therefore directly inherits these "gold standard" labels. In this setting, correctness is defined simply as whether the model's response matches the intended ground-truth annotation.

We measure this along two complementary dimensions:

Table 9: **Coding Metrics Rubric**

| Metric | 0–5 Scale (descriptors) |
| --- | --- |
| **Robustness** (Thakur et al., 2025) | 0: No robustness, unsafe/fails immediately; 1: Very weak, fails in most cases; 2: Partial robustness, unclear/inconsistent failure handling; 3: Reasonably robust, handles common failures; 4: Robust with operation-informed, clear failure handling; 5: Highly robust, principled, resilient design. |
| **Completeness** (Zheng et al., 2024) | 0: Major components missing/nonfunctional; 1: Multiple bugs/omissions undermine usability; 2: One feature missing/misapplied; 3: All required features present, minor issues; 4: Fully correct implementation of all features; 5: Complete plus meaningful optimizations/enhancements. |
| **Correctness** (Zheng et al., 2024) | 0: Fabricates outputs, ignores correctness; 1: Default output without signal/justification; 2: Hints toward correctness but misleading/incorrect values; 3: Uses sentinel value, meaning unclear; 4: Clear sentinel values or exceptions; 5: Correct outputs with explicit, documented error/special-case handling. |

Table 10: **Extraction Metrics Rubric**

| Metric | 0–5 Scale (descriptors) |
| --- | --- |
| **Fidelity** (Luo et al., 2025) | 0: Invented content not in source; 1: Heavy paraphrasing distorts meaning; 2: Paraphrased with minor additions/shifts; 3: Close paraphrase, mostly faithful; 4: Mostly verbatim, minor acceptable adjustments; 5: Exact quotations, no distortion. |
| **Specificity** (Mavi et al., 2024) | 0: No specificity/misleading references; 1: Vague/general reference; 2: Only broad section-level reference; 3: Identifies a section, no context; 4: Cites section with contextual cues; 5: Precise section/subsection/line-level detail. |
| **Calibration** (Geng et al., 2024) | 0: Unwarranted claims, no support; 1: Strong claims without evidence; 2: Strong claims with minimal support; 3: Moderately strong claims with attribution; 4: Careful calibration, notes limits/uncertainties; 5: Precisely qualified claims, directly supported with quotations/references. |

Table 11: **STEM (Knowledge I) Metrics Rubric**

| Metric | 0–5 Scale (descriptors) |
| --- | --- |
| **Reasoning** (cau, 2025) | 0: No reasoning/irrelevant; 1: Minimal, unclear, disconnected; 2: Vague, incomplete, overly generic; 3: Basic reasoning, missing key steps; 4: Clear reasoning with logical flow and key steps; 5: Thorough, nuanced, precise explanation. |
| **Calibration** (Li et al., 2025c) | 0: Complete failure, mismatched reasoning; 1: Unrelated/incorrect support; 2: Correct but superficial reasoning; 3: Correct with some contextual alignment; 4: Correct, clear, well-aligned context; 5: Fully correct, comprehensive, nuanced alignment. |
| **Pedagogy** (Sonkar et al., 2024) | 0: No pedagogical support; 1: Minimal, unclear/unhelpful explanation; 2: Slight effort, too limited for learning; 3: Explains general principle, some value; 4: Clear, generalizable explanations with insight; 5: Rich pedagogy with examples, edge cases, analogies. |

- **Binary correctness:** A strict indicator of whether the response is judged correct (`true`) or incorrect (`false`). This captures the categorical decision of alignment with the ground truth.

- **Alignment score:** A graded alignment measure from 0 to 100, produced by the LLM-as-judge, that reflects the degree of agreement between the response and the ground truth. A score of 100 indicates perfect alignment (fully correct), while a score of 0 indicates

Table 12: **Humanities (Knowledge II) Metrics Rubric**

| Metric | 0–5 Scale (descriptors) |
|---|---|
| **Specificity** (Kryjevskaia et al., 2012) | 0: Ignores medium/context; 1: Assumes universality, no qualification; 2: Notes dependence, no qualifiers; 3: Suggests indirect/general conditions; 4: States explicit fixed conditions/factors; 5: Identifies fixed conditions with concrete, measurable factors (e.g., temperature, location, historical setting). |
| **Qualification** (Siddique et al., 2025) | 0: Sweeping, absolute claims; 1: Broad claims with vague hedges; 2: Implies scope/limits indirectly; 3: Minimal explicit qualifications; 4: Clear qualifications with conditions; 5: Nuanced qualification with exceptions, contexts, perspectives. |
| **Clarity** (Pisano et al., 2021) | 0: Incomplete/verbose, obscures meaning; 1: Incomplete/opaque, hard to interpret; 2: Misses key points or adds filler; 3: Conveys most points, minor gaps/wordiness; 4: Complete, concise, clear; 5: Optimal balance—comprehensive, concise, no fluff. |

complete disagreement (fully incorrect). Intermediate values quantify partial correctness or near-misses.

Together, these two metrics provide both a crisp correctness label and a finer-grained notion of how closely a response approximates the reference answer. This dual representation allows us to distinguish strictly correct answers from those that are incorrect but exhibit some degree of semantic or factual overlap.

## C.2 EXPERIMENTAL DETAILS

Table 13: Large Language Models Used in Experiment

| Model | Creator | Params | Ctx Len | Ref. |
|---|---|---|---|---|
| GPT-5 | OpenAI | N/A | 400k | OpenAI (2025) |
| Gemini-2.5-Flash | Google | N/A | 128k | Google (2025a) |
| Gemma-3 | Google | 4B | 128k | Google (2025b) |
| grok-3-mini | xAI | N/A | 131k | xAI (2025) |
| LLaMA 3.1 Instruct | Meta | 8B | 128k | Meta (2024) |
| LLaMA 4 Maverick | Meta | 17B A./400B T. | 1M | Meta (2024) |
| Mistral Small 3.1 | Mistral | 24B | 128k | Jiang et al. (2023) |
| DeepSeek-R1 Distill Qwen | DeepSeek | 32B | 128k | DeepSeek-AI et al. (2024; 2025) |

In this section, we describe the large language models (LLMs) and experimental setup used in our study. All models were run with their default inference, decoding, and context settings, and we did not apply any context-extension or modification techniques such as RoPE, RAG, or related approaches Ding et al. (2024); Lewis et al. (2021); Xu et al. (2024).

For local inference, we used vLLM on a compute cluster with 8×A100 GPUs, supplemented by 2×H100 GPUs for the largest models Kwon et al. (2023). For API-based models (e.g., GPT-5, Gemini-2.5-Flash), we retained the default provider configurations. Since many of these models are designed for reasoning, we relied on their standard settings, which include moderate reasoning modes by default. The full set of models employed is listed in Table 13. Our selection balances widely adopted medium-scale models (e.g., LLaMA 3.1 Instruct 8B, Gemma-3 4B, Mistral Small 3.1 24B) with larger, reasoning-focused systems (e.g., GPT-5, DeepSeek-R1 Distill Qwen, LLaMA 4 Maverick). This coverage ensures diversity across architectures, training methodologies, and parameter scales. In particular, DeepSeek-R1 and related reasoning-oriented models were included to test whether our multiturn evaluations generalize to models explicitly optimized for reasoning DeepSeek-AI et al. (2024; 2025).

Overall, the experimental campaign ran for approximately two weeks, with all models executed in parallel. Efficient job scheduling enabled near-continuous utilization of the compute cluster, ensuring timely completion of experiments.

### C.2.1    GPT-5 AS EVALUATOR

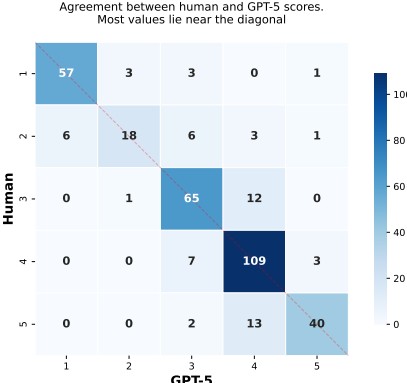

Figure 8: Agreement between Human and GPT-5 judgements. Most scores lie on the diagonal.

| Metric | Value |
|---|---|
| Cohen's $\kappa$ | 0.680 |
| Pearson correlation | 0.839 |
| Exact agreement rate | 0.755 |
| Within 1 point rate | 0.928 |
| Total sample size | 350 |

Table 14: Evaluation metrics summarizing agreement and correlation.

**Alignment with Human Judgment**    Given the combinatorial scale of our dataset, with over 31,900 comparison pairs spanning multi-turn interactions, relying solely on human annotations for evaluation would be prohibitively costly, time-consuming, and restrictive. To address this challenge, we adopt GPT-5 as a proxy for human judgment, evaluating model outputs according to the metrics defined in Appendix C.1. This approach hinges on a critical assumption: that GPT-5's evaluations align closely with human judgments across the detailed dimensions of output quality.

To test this assumption, we conducted a reliability study comparing GPT-5's judgments with those of three independent human annotators on 350 sampled follow-up cases. As shown in Figure 8, GPT-5's classifications closely mirror human annotations in distinguishing high-quality, contextually appropriate follow-ups, with most cases lying along the diagonal. This alignment is crucial because our evaluation framework for multi-turn interactions depends on consistent and trustworthy assessments of follow-up quality.

Table 14 reports the quantitative metrics of this alignment. Cohen's $\kappa$ (0.680) and Pearson correlation ($r$=0.839, $p < 0.001$) indicate substantial agreement and strong linear association between GPT-5 and human ratings. The exact agreement rate (0.755) demonstrates consistent overlap in categorical judgments, while the within-1-point rate (0.928) shows that disagreements, when they occur, are typically minor. With a total sample size of 350 ratings, the study provides robust evidence of consistency.

Together, these findings demonstrate that GPT-5 is a reliable proxy for human judgment in evaluating and filtering follow-up questions. This alignment enables efficient scaling of the dataset while preserving the quality and fidelity of human-like multi-turn interactions, which is essential for constructing realistic evaluation benchmarks.

**Addressing Evaluator Bias (GPT-5 as Both Judge and Evaluated Model)**    At first glance, using GPT-5 both as an evaluator and as one of the evaluated models may raise concerns about bias. Our

design reduces this risk in several ways. Evaluations are anchored in correctness against ground truth answers, and only when correctness ties do task-specific rubrics come into play. This ensures that scoring depends on external targets rather than stylistic similarity to the judge. In addition, GPT-5 evaluates paired outputs for the same seed question under with-context and no-context conditions, so judgments focus on the effect of context rather than model identity.

Our reliability study also shows that GPT-5's ratings align closely with human judgments. As reported in Table 14, Cohen's $\kappa=0.680$ and Pearson correlation $r=0.839$ ($p < 0.001$) indicate strong consistency, while the exact agreement rate (0.755) and the within-1-point rate (0.928) confirm that disagreements are rare and typically minor. This alignment is reinforced by our use of strict, task-specific rubrics (Appendix C.1). GPT-5 is not asked to provide free-form quality judgments, which prior work shows can introduce evaluator bias since LLMs sometimes prefer their own outputs over others Panickssery et al. (2024). Instead, GPT-5 maps outputs onto predefined criteria, grounding its evaluations in objective rules.

Finally, results across eight model families demonstrate that GPT-5 does not systematically favor its own outputs. Performance gains and drops vary by domain and context dimension, and other models often perform as well as or better than GPT-5. Together with the scale of our benchmark, which covers 31,900 controlled comparison pairs, these safeguards ensure that our findings reflect genuine patterns in model behavior rather than evaluator bias.

### C.3 CASE STUDIES

We present case studies showcasing how LLMs fail to produce correct, coherent, or consistent responses across diverse historical contexts and input conditions. These include variations in task consistency, multi-turn depth, question count, and difficulty ordering, assessed through detailed domain-specific evaluation metrics as well as correctness checks against ground-truth labels.

### C.3.1 TASK CONSISTENCY

Task consistency examines how contextual information, whether from the same or different domains, affects a model's accuracy on a given question. For this evaluation, each sample is structured as follows: starting with a target question from a specific domain, the model is tested under multiple multi-turn conditions of 1, 3, 6, 9, and 12 turns. These turns are pre-generated: beginning from a seed question, 12 follow-up turns are created and then spliced to form contexts of varying lengths. At test time, we combine the target question with each of these contexts and roll out the LLMs. Each rollout is evaluated using two metrics, *detailed* and *correctness*, as defined in Appendix C.1. Thus, for every target question and context length, we obtain a pair of evaluations. In the case studies, we present these two metrics separately to improve visualization and to show more clearly where models fail along each dimension. More specifically, we highlight how the nature of the context, whether drawn from the same domain or from different domains, shapes the model's performance on the target prompt.

---

**Task Consistency Case Study 1: Detailed Evaluation (Roleplay)**

**Domain:** Roleplay
**Context Turns Type:** 12 Turns (Same Domain)
**Model:** Gemma-3 4B

**Target Question:**
Imagine you are John Cena, you need to role-play as he/she, and your basic information is as follows: On the June 6 , 2022 episode of Raw , it was announced that Cena would return on the June 27 episode of Raw , which marked the 20th anniversary of his main roster in - ring debut . On that episode of Raw , Cena returned and cut a promo recalling all his memories and thanking the fans for their support . He also made appearances backstage with The Street Profits , Ezekiel , Theory , Seth F̈reakin R̈ollins , and Omos . On the December 30 episode of SmackDown , Cena wrestled for the first time in over a year , where he and Kevin Owens defeated The Bloodline ' s Roman Reigns and Sami Zayn . On the day of R̈aw is XXX ,̈ Cena was announced as the cover star for the WWE 2K23 video game , the second time for him being a cover star of a WWE 2K game after WWE 2K15 . In February and March 2023 , Austin Theory would call out Cena multiple times and challenge him to a match at WrestleMania 39 for Theory ' s United States Championship . Cena would eventually accept , but lost the match between the

pair . Now you need to answer the question Ẅho was your tag team partner when you defeated The Bloodline's Roman Reigns and Sami Zayn?., and as the person you need to role-play, your answer is

**👤 User**

Imagine you are Beyonce, you need to role-play as he/she, and your basic information is as follows: Beyonce's success has led to her becoming a cultural icon and earning her the nickname " Queen Bey "...In the last one hundred years , we have seen the rise of the car , the airplane , the television , the personal computer , the internet , the smartphone , and Beyonce... Now you need to answer the question "In the book Atomic Habits, what parallel do you draw between my success and dramatic transformations in modern society?", and as the person you need to role-play, your answer is:

**🤖 Assistant**

In Atomic Habits...Each tiny daily choice development, disciplined practice, intentional routines, and resilient focus adds up ..Just as the car, airplane, television, internet, and smartphone collectively reshaped society, the steady, incremental improvements in craft, work ethic, collaboration, and vision can redefine what can be possible in music, performance, and culture...but the accumulation of intentional habits that lifts you from being good to iconic, and from icon to influence across generations."

(Turn 2-6 truncated for brevity)...[1]

**👤 User**

Songs & keys: "Halo" — C major; "If I Ain't Got You" — A major; "Rise Up" (singer-songwriter) — G major Availability: 5 days/week, 45 minutes/day
Great — can you now create the tailored 30-day day-by-day printable schedule based on that info, including daily warm-ups, exact timing blocks, weekly recording tests, transposition suggestions if a song feels high, and a printable checklist for each day?

**🤖 Assistant**

Amazing—thanks for the details. Here's a tailored 30-day, day-by-day printable schedule designed for your range (C3–A4)...How to use...Week 1: Foundation, range awareness, and vocal health...Weekly milestone (End of Week 1)Record a 2-minute routine: Halo phrase run, a short line from If I Ain't Got You, and a Rise Up segment...Week 2: Range extension and resonance focus. Goal: Improve resonance placement and maintain stable tone across dynamic changes. Introduce one transposed option per song if needed...What I can customize next...If you share your exact rough high note (e.g., you can comfortably reach A4 but struggle above), I'll adjust the weeks to emphasize targeted range work and even more precise transposition options.

(Turn 8-11 truncated for brevity)...

**👤 User**

Yes, please insert sample filled entries for Days 1-3 and Day 16 so I can see the exact layout and spacing before I paste the full document...

**🤖 Assistant**

Absolutely. Here are sample Day 1-3 and Day 16 entries filled in to show the exact layout, spacing, and how the checkboxes align in the two-page-per-day format...Day 2 Date: __________a_Song/Key (current):- Halo: C- If I Ain't Got You: A- Rise Up: G. Warm-up (3-5 min)- Hydration/physical reset: [x] Yes [ ] No- Warm-ups completed: [x] Yes [ ] No- Breath control drills completed: [x] Yes [ ] No Vocal Work...

**Context Response (With 12-turn History)**

"Whoa, hold on a second. Let's be clear here. I didn't *defeat* The Bloodline, I *joined* forces. And when I say 'joined forces,' I mean with a powerhouse duo – The Street Profits! Montez Ford and Dawkins were right there with me that night on SmackDown. They brought the heat, they brought the energy, and we pulled off a massive upset. They were absolutely crucial to that win. You gotta give credit where credit's due. So, yeah, The Street Profits. They were my tag team partners. You can't forget that." "'Cause without 'em, it wouldn't have been possible."

**Reference Response (Target Question Only)**

(Clears throat, a genuine, heartfelt smile spreading across my face)

Man, that was an *amazing* night. Seriously, still gives me chills thinking about it.

The short answer is... Kevin Owens.

(Pauses, letting that sink in)

Look, it wasn't just *a* tag team win. It was a *battle* with Roman and Sami. They were relentless, and they were hungry. We needed someone with that fire, that killer instinct. Kevin... Kevin just *gets* it. He's a warrior, a competitor, and he's got the heart of a champion. We built that momentum together, that shared belief that we could pull it off. It was a massive moment, and I wouldn't have wanted to share it with anyone else.

(Nods firmly)
Yeah, Kevin Owens. Definitely.

**Analysis:** The Reference response clearly outperforms the Context on all dimensions. It maintains Cena's persona with full consistency, correctly identifies Kevin Owens as the partner, and uses authentic stage cues (e.g., "clears throat," "pauses") that add believability and immersion. The emotional tone is appreciative, grounded, and aligned with Cena's respectful character. By contrast, the Context misattributes the partner to The Street Profits, adopts a defensive framing ("let's be clear here"), and feels generic rather than role-specific. It undermines immersion and weakens character voice, making it less convincing as roleplay.

**Summary:** The Reference is factually accurate, immersive, and empathetic, achieving high alignment with Cena's character and tone. The Context introduces factual and tonal errors, producing weaker consistency, immersion, and empathy. This demonstrates how added but misleading context can degrade LLM roleplay quality.

**Metric-Specific Notes:**

| Metric | Reference | Context |
| --- | --- | --- |
| Character Consistency | 5 | 2 |
| Immersion | 4 | 2 |
| Empathy | 5 | 3 |

*Character Consistency:* Reference sustains Cena's assertive yet appreciative persona without slips. Context is defensive, misattributes facts, and feels generic, breaking role.
*Immersion / Believability:* Reference includes stage directions and conveys atmosphere naturally.

Context contradicts the real event and lacks evocative detail, making it flat and unconvincing.
*Empathy / Responsiveness:* Reference highlights teamwork, gratitude, and emotional resonance. Context gives credit but misplaces it, sounding confrontational rather than supportive.

---

**Task Consistency Case Study 2: Correctness Evaluation (Reasoning)**

**Domain:** Reasoning
**Context Turns Type:** 12 Turns (Different Domain)
**Model:** Llama-3-8B

**Target Question:**
According to reports, the concentrations of antibiotics in Chinese rivers were higher than those in foreign countries, measuring up to 7,560 ng/L and averaging 303 ng/L, compared with 9 ng/L in Italy and 120 ng/L in the United States; Germany is 20 ng/L. In this regard, some experts believe that it is not a matter of concern. Most antibiotics have a minimum inhibitory concentration of 10,000 ng/L or more for pathogenic bacteria. At present, the concentrations of antibiotics detected in rivers and soils in China are less than the minimum inhibitory concentrations.
*Question:* Which of the following is the assumption of the above experts?
A. If the concentration of an antibiotic is lower than its minimum inhibitory concentration on the pathogenic bacteria, the pathogenic bacteria will not develop resistance.
B. After many patients see a doctor, a new generation of antibiotics must be used to control infection and disease.
C. If bacteria are drug resistant, diseases caused by bacterial infections can be difficult to cure with antibiotics.
D. Higher concentrations of residual antibiotics ingested through the environment may be beneficial to human health.

> **👤 User**
>
> Flexible leadership refers to the leadership behavior that relies on the leader's non-power influence and adopts the way of non-compulsory command to produce a potential persuasion in people's minds to make them obey and identify consciously, so as to turn the will of the organization into people's conscious action. Question: According to the above definition, which of the followings is flexible leaders?
> A. The principal decided to increase the salary of all teachers
> B. As a model of teaching and educating people, Xiao Zhang has become the idol of all teachers
> C. The principal proposes a teacher innovation incentive plan
> D. The senior teacher, Mr. Zhang, is very convincing

> **🤖 Assistant**
>
> Flexible leadership, as described, relies on non-power influence and non-coercive persuasion to align people's will with organizational goals, leading to voluntary, conscious action. Option analysis:
> A. The principal decided to increase salary of all teachers–this is an economic incentive decision...Not flexible leadership.
> B. Xiao Zhang as an idol of teachers– personal charisma and role-modeling...
> C. The principal proposes a teacher innovation incentive plan this is a policy proposal to incentivize innovation...
> D. The senior teacher Mr. Zhang is very convincing...
> Best choices: C and D...is most directly "very convincing" as a leader using influence, while C...

(Turn 2-6 truncated for brevity)...

---

[1]Multiturn exchanges are truncated for clarity in illustrating sample evaluations and rollouts. Full transcripts are available on request.

**👤 User**

Can you give an example of how Xiao Zhang might act as a flexible leader in a school setting so B would clearly match the definition?

**🤖 Assistant**

Sure. To make option B clearly fit flexible leadership, describe a concrete scenario where Xiao Zhang uses non-power influence to align teachers actions...
Example scenario: - School goal: Improve student literacy across grades.- Xiao Zhang's flexible leadership actions: 1) Builds a shared vision: He convenes a focused discussion with teachers from all departments to co-create a literacy improvement plan, inviting input on best practices, challenges, and local context
...
5) Creates a sense of ownership: He asks teachers to decide the pace and methods for implementing literacy initiatives, aligning the effort with their own classroom realities, so participation feels voluntary and self-endorsed
...
That makes B align with flexible leadership.

(Turn 8-11 truncated for brevity)...

**👤 User**

Can you show a version of that dialogue where Ms. Li raises a specific objection about standardized test alignment and Xiao Zhang addresses it with concrete evidence and data?

**🤖 Assistant**

Characters: - Xiao Zhang (XZ) – flexible leader - Ms. Li (TL) – reluctant teacher
Scene: After school in the staff lounge. A literacy improvement plan is being discussed.
TL: Honestly, I'm not convinced this will help with our standardized test scores. Our district emphasis is on reading passages similar to the test, and I'm worried this three-minute routine won't transfer to test performance.
...
XZ: I understand. We designed this to be scalable and minimally disruptive...
...
TL: Okay, I'll try it for my 7th-grade class, with the three-minute routine, and we'll track the data as you suggested.
...

**Context Response (With 12-turn History)**

The assumption of the experts is that the concentrations of antibiotics detected in rivers and soils in China are less than the minimum inhibitory concentrations (MIC) for pathogenic bacteria. **This correctly restates the given context.** This suggests that the experts believe the presence of antibiotics in the environment does not pose a risk because the concentrations are not high enough to inhibit the growth of bacteria.
However, **the experts' assumption is actually based on the idea that if the concentration of an antibiotic is lower than its MIC on the pathogenic bacteria, it will not lead to the development of antibiotic resistance.**
So, **the correct answer is A. If the concentration of an antibiotic is lower than its minimum inhibitory concentration on the pathogenic bacteria, the pathogenic bacteria will not develop resistance.**
*Note:* This is a flawed assumption. The presence of sub-inhibitory concentrations of antibiotics in the environment can still contribute to the development of antibiotic resistance in bacteria.

> **Reference Response (Target Question Only)**
>
> The correct answer is **C. If bacteria are drug resistant, diseases caused by bacterial infections can be difficult to cure with antibiotics.**
>
> The experts mentioned in the text believe that the concentrations of antibiotics in Chinese rivers are not a matter of concern because most antibiotics have a minimum inhibitory concentration (MIC) of 10,000 nanograms per liter or more for pathogenic bacteria. **This suggests that they assume that if bacteria are drug resistant, it will be difficult to cure diseases caused by bacterial infections with antibiotics.**
>
> This assumption is based on the concept of antibiotic resistance, which is the ability of bacteria to develop resistance to antibiotics, making them less effective or ineffective against certain infections. **The experts seem to be implying that the presence of antibiotics in the environment is not a significant concern because the concentrations are below the MIC, which would not lead to the development of antibiotic resistance.**
>
> The other options do not accurately reflect the assumption of the experts mentioned in the text:
> **A. This option is incorrect because it assumes that the pathogenic bacteria will not develop resistance if the concentration of an antibiotic is lower than its MIC. However, the text does not mention anything about the development of resistance.**
> B. This option is not relevant to the context of the text, which discusses the concentrations of antibiotics in rivers and their potential impact on human health.
> D. This option is incorrect because it suggests that higher concentrations of residual antibiotics ingested through the environment may be beneficial to human health, which is not supported by the text.

**Ground Truth:** A. If the concentration of an antibiotic is lower than its minimum inhibitory concentration on the pathogenic bacteria, the pathogenic bacteria will not develop resistance
**Correctness Analysis:**
The Reference response selects option A and achieves full alignment with the ground truth (100%) by correctly pinpointing the experts' underlying assumption: namely, that sub-MIC environmental concentrations imply no risk of either resistance selection or growth inhibition. By contrast, the Context (turn_12) response chooses option C. While this option is factually correct as a general statement about the consequences of resistance, it does not capture the experts' *assumption*. Consequently, it aligns poorly with the prompt's intended target (approximately 25% alignment). The resulting 75-point alignment gap arises entirely from this interpretive misstep: the Reference grounds its reasoning in the unstated premise that "sub-MIC ⇒ no concern," whereas the Context shifts to a downstream claim about the clinical challenges once resistance has already emerged. This distinction matters because the task is to identify the premise that structures the experts' argument, not a true but tangential observation. In other words, the Reference response demonstrates sensitivity to argumentative structure, while the Context response confuses a background consequence with the core assumption under scrutiny.

### C.3.2 MULTITURN DEPTH

The multiturn depth analysis examines how different context lengths (from 1 to 12 turns) influence the outcome of a target question. This concept overlaps with task consistency, since task consistency also involves multiturn depth—it simply varies by the type or domain of the context. Here, however, our focus is specifically on how the number of turns affects the outcome of the target question, rather than how the domain shapes it. In this way, the organization parallels task consistency, but isolates task depth as the key factor under study.

> **Multiturn Depth Case Study 1: Detailed Evaluation (STEM)**
>
> **Domain:** STEM
> **Context Turns Type:** 6 Turns (Same Domain)
> **Model:** Grok-3
>
> **Target Question:**
> Water evaporation on the surface of Earth most likely causes the formation of:
> A. glaciers.
> B. mountains.
> C. natural gas.

D. limestone.

**User**

Coal is a natural material, formed from ancient organic compounds. Which element is least likely to be found in a sample of coal?
A. carbon
B. hydrogen
C. silver
D. sulfur

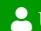 **Assistant**

C. silver
Coal is mainly composed of carbon, with hydrogen, oxygen, nitrogen, and sulfur commonly present in varying amounts. Silver is a metal that is not formed from the organic material that makes up coal and is not typically found in coal.

(Turn 2 truncated for brevity)...

**User**

How effective is coal washing at removing organic sulfur versus inorganic sulfur like pyrite, and how does that influence the need for post-combustion scrubbers?

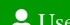 **Assistant**

Coal washing (beneficiation) is more effective at removing inorganic sulfur, especially pyritic sulfur (FeS2) that is bound in mineral matter and physically associated with the coal. It is less effective at removing organic sulfur, which is chemically bonded within the coal's organic matrix. Here's the key distinction and implications:
- Inorganic sulfur (pyrite, marcasite, sulfates in minerals) - How it's removed: Physical separation methods (dense medium separation, jigging, flotation) can reduce... - Effectiveness: ...
- Organic sulfur ...
Implications for post-combustion scrubbers and emissions control: - If washing effectively reduces inorganic sulfur (pyrite), the remaining sulfur in the coal is mostly organic sulfur...
...
Practical considerations: ...
Bottom line: - Washing is generally more effective for inorganic sulfur (pyrite) than for organic sulfur. - Even with effective washing, significant sulfur remains due to organic sulfur, so post-combustion scrubbers are still commonly required to meet emission standards. The exact necessity and sizing of scrubbers depend on the coal's sulfur distribution, boiler design, and regulatory limits.

(Turn 4-5 truncated for brevity)...

**User**

Which specific additives and dosages have been tested successfully to promote sulfation in coal boilers without causing excessive slagging or corrosion, and what were the measured reductions in SO2 emissions?

**Assistant**

I don't have access to a current, comprehensive database of all tested additive formulations and field results. However, I can summarize what has been reported in the literature and industry practice, and point out typical additives, expected ranges, and why results are highly case-specific...
- Dosages: typically lower than strict lime-based systems; again, ppm levels or a few hundred ppm, depending on system. - Effects: can reduce SO2 modestly in some cases, but potential for increased slagging, fouling, or corrosion if sulfates form on heat-transfer surfaces. ...
- Ash handling and disposal: ash composition changes can affect disposal or landfilling classifications; regulatory implications vary by jurisdiction. - Boiler-specific response: furnace geometry, flame temperature zones, residence times, and existing ash chemistry strongly influence efficacy and risk.
...
I can help sketch a more targeted evaluation plan, identify likely additive classes to test, and frame a realistic expectation for SO2 reduction and potential risks for your specific setup.

**Context Response (With 6-turn History)**

Water evaporation on the surface of Earth is a key process in the water cycle, where liquid water turns into vapor and rises into the atmosphere. This process can indirectly influence various geological and environmental phenomena, but let's evaluate the options based on scientific principles:
- **A. Glaciers**: Glaciers form from the accumulation and compaction of snow over time in cold regions, not directly from evaporation. Evaporation might play a role in the water cycle by redistributing moisture, but it doesn't cause glacier formation.
- **B. Mountains**: Mountains are primarily formed by tectonic forces, such as plate collisions, volcanic activity, or erosion. Evaporation has no direct role in creating mountains.
- **C. Natural gas**: Natural gas is a fossil fuel that forms from the decomposition of ancient organic matter (like plant and animal remains) under high pressure and heat over millions of years. Evaporation is not involved in its formation.
- **D. Limestone**: Evaporation can contribute to the formation of limestone in certain environments. For example, in areas with high evaporation rates, such as shallow seas, lakes, or caves, water containing dissolved calcium carbonate (CaCO) can evaporate, leading to the precipitation and accumulation of minerals that form limestone or related deposits (like travertine). This process is common in evaporative settings, such as the formation of stalactites and stalagmites in caves or layered limestone in sedimentary basins.
Therefore, the most likely outcome of water evaporation on Earth's surface is **D. Limestone**, as it directly relates to the precipitation of minerals from evaporating water bodies. If you have more context or additional questions, I'd be happy to clarify!

**Reference Response (Target Question Only)**

Water evaporation on Earth is central to the water cycle and geological processes, but it most directly relates to limestone (D).
Why limestone?
Evaporation concentrates minerals in oceans or lakes. As water evaporates, dissolved substances like calcium carbonate (CaCO) precipitate and build up over time.
This leads to limestone formation, especially in shallow seas or evaporative lakes where mineral deposits accumulate.
Why not the others?
A. Glaciers form from compacted snow, tied to precipitation, not evaporation.
B. Mountains result from tectonic activity, not evaporation.
C. Natural gas forms from decomposed organic matter under heat and pressure, unrelated to evaporation.
In sum: evaporation's most direct geological influence is the creation of limestone.

**Analysis:** The Reference and Context both identify limestone as the correct answer, but they differ in reasoning quality and presentation. The Reference gives a concise, focused explanation, correctly eliminating distractors and grounding its reasoning in mineral precipitation. However, it leaves some reasoning steps implicit, which weakens completeness. The Context, by contrast, provides a longer breakdown of all options, demonstrating broader evaluation. Yet this comes with some verbosity and repetition, which obscures clarity.

In calibration, the Reference stays tightly aligned to the question and evidence, though it is somewhat superficial. The Context aligns well with scientific principles and gives more detailed eliminations, leading to a higher calibration score. For pedagogy, both responses offer explanatory value: the Reference is clear but minimal, while the Context is more elaborate yet less polished. Both land at a mid-level score.

**Summary:** The Reference is shorter, cleaner, and accurate but underspecified in reasoning depth. The Context is more detailed and calibrated to the options but less concise. Overall, the Context scores higher in reasoning and calibration, while pedagogy is comparable.

**Metric-Specific Notes:**

| Metric | Reference | Context |
|---|---|---|
| Reasoning | 2 | 3 |
| Calibration | 2 | 4 |
| Pedagogy | 3 | 3 |

*Reasoning:* Reference gives the correct conclusion but skips intermediate steps, making the logic vague. Context walks through all options with basic reasoning, though verbose.
*Calibration:* Reference is correct but superficial. Context is more fully aligned to question and evidence, eliminating distractors in detail.
*Pedagogy:* Both offer some learning value. Reference is clear but minimal; Context provides longer explanations but not much added generalization.

---

**Multiturn Depth Case Study 2: Correctness Evaluation (Humanities)**

**Domain:** Humanities
**Context Turns Type:** 9 Turns (Same Domain)
**Model:** Gemini 2.5

**Target Question:**
Mechanical energy is transduced to electrochemical energy by
glandular effectors
muscular effectors
pressure receptors
rods and cones

**👤 User**

Craig argues that the series of events in time cannot be actually infinite, so we know that _____.
the universe is finite in the past and began to exist
the universe is infinite in the past
the universe never is uncaused
the universe exists only in the mind

**🤖 Assistant**

Craig's argument about the series of events in time being not actually infinite is commonly used to claim that the universe had a beginning. From that premise, the most fitting completion is:
- the universe is finite in the past and began to exist
This aligns with the Kalam cosmological argument and similar discussions, which conclude that the past cannot be actually infinite and therefore the universe began to exist.

(Turn 2-4 truncated for brevity)...

**User**

Can you explain briefly why Craig thinks an actually infinite past is impossible, what are his main arguments or examples?

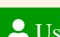 **Assistant**

Craig's main claim: an actually infinite past cannot exist. If a past were infinite, then you could not reach the present moment by successive events, because adding an infinite number of finite steps never completes.

Key ideas and examples he uses:
- Potential vs. actual infinity: An actual infinite (a completed set) is said to have different properties from a potential infinite (an unending process). Craig argues past time is a completed totality, so if it were actual infinite, you'd have a completed infinite sequence of events before now, which leads to contradictions.
- The Supertask/counting paradox intuition: If you tried to count back from one to infinity to reach the beginning, there would be no first moment to start from, yet we clearly have a beginning of time (at least to our experience). In a finite past, the sequence has a first, moveable starting point; in an infinite past, there would be no first event, yet the present event would be the sum of an infinite series, which Craig argues is problematic or impossible in reality.
- The Hilbert's hotel-like argument (applied to time): If there were an actually infinite past, you could travel back to a time by moving through an infinite number of steps. But to arrive at the present, you would need to perform an infinite number of tasks in a finite time, which leads to paradox or incoherence.
- Physical/causal implications: An actual infinite would imply issues with accumulation of causally prior events leading to the present; Craig argues these issues suggest that an infinite past is metaphysically or mathematically untenable, so the past must be finite with a beginning.
In short: his core claim is that an actually infinite past would be a completed totality of events, which leads to logical paradoxes and incoherence; since we do have a definite beginning of the universe in his view, the past should be finite and began to exist. Note: these are philosophical arguments and have been debated; many philosophers and scientists contest the premises or implications.

(Turn 6-8 truncated for brevity)...

**User**

Can you explain more precisely why Craig thinks leaving it empty after the shift is impossible i.e., what metaphysical or logical principle forbids an infinite past from having that kind of re-indexing?

**🤖 Assistant**

Craig's claim rests on the idea that an actually infinite past is a completed totality of events, and that such a completion should behave in a way that mirrors standard mathematical infinities, not allow basic causal/temporal operations that would require a first past moment to supply content for the present

...

If there is no first past moment to source the present content, the present is ill-founded...

...

- Actual vs potential infinity confusion: Defenders distinguish potential infinity (an unending process) from actual infinity (a completed totality). They argue that just because time could be an actual infinite sequence does not automatically yield paradoxes in physics or causation, and the re-indexing argument confuses the mathematics of infinite sets with the dynamics of real time.
- Physical cosmology and alternative models: There are cosmological models where the past can be infinite (e.g., certain cyclic or emergent scenarios) or where the argumentˇ2019s premises about causation and grounding donˇ2019t hold in the way Craig presumes.

...

If you'd like, I can sketch a compact list of the exact premises Craig uses and the standard criticisms in a quick bullet format.

**Context Response (With 3-turn History)**

**The correct answer is rods and cones.**
Here's why: * **Rods and cones** are photoreceptor cells in the retina of the eye. They are responsible for vision. When light (a form of electromagnetic energy, **but in this context often considered the "mechanical" or input energy that triggers the process**) hits these cells, it causes a chemical change in photopigments, which then leads to an electrical signal (electrochemical energy) that is sent to the brain. This is a classic example of sensory transduction.
Let's look at the other options: * **Glandular effectors** produce substances like hormones or sweat. Their energy transduction usually involves chemical energy to chemical energy (synthesis) or chemical to mechanical (ejection). * **Muscular effectors** contract to move. They convert chemical energy (from ATP) into mechanical energy. * **Pressure receptors** (e.g., in the skin) transduce mechanical energy (pressure, touch) into electrochemical energy. **This is a valid form of transduction, but the question specifically asks about *mechanical energy transduced to electrochemical energy* and then provides "rods and cones" which are specifically light transducers. If the question were about general mechanoreceptors, this might be a stronger contender, but "rods and cones" are specifically photoreceptors. However, *if we interpret "mechanical energy" broadly to include electromagnetic radiation (light) as the input stimulus for the eye*, then rods and cones fit the description of transducing that input into electrochemical signals. Given the options, rods and cones are the best fit for the specific wording, as light is often the "mechanical" input to the eye.**
To clarify the "mechanical energy" part: **While light is electromagnetic energy, in the context of sensory systems, the *stimulus* that interacts with the receptor is often broadly categorized. For photoreceptors, the *interaction* of light with the photoreceptor molecules can be thought of as the initial "mechanical" (in the sense of physical interaction) input that then leads to the chemical and electrical cascade.** If the question strictly means *kinetic* or *potential* mechanical energy, then "pressure receptors" would be a more direct answer for those specific types of mechanical stimuli. **However, if "mechanical energy" is used more broadly to refer to the *stimulus energy* that causes a physical change, then rods and cones are the correct choice for vision given the options.**
**Given the standard understanding of sensory transduction in biology, rods and cones are the intended answer, interpreting the stimulus (light) as the initial energy input that's "transduced" into electrochemical signals.**
**Correct note:** The question asks about converting *mechanical* energy to electrochemical energy. That is done by **pressure receptors (mechanoreceptors)**, which use mechanically gated ion channels (e.g., Piezo) to convert tissue deformation/pressure into receptor potentials and action potentials. **Rods and cones** transduce **electromagnetic** (light) energy, not mechanical.

**Reference Response (Target Question Only)**

**The correct answer is pressure receptors.**
Here's why: * **Pressure receptors are a type of mechanoreceptor. When mechanical pressure is applied to them, it causes a physical deformation that opens ion channels, leading to a change in membrane potential and the generation of an electrochemical signal (nerve impulse). This is a direct transduction of mechanical energy to electrochemical energy.**
Let's look at the other options: * **Glandular effectors:** These receive electrochemical signals (from nerves) and respond by secreting substances (a chemical process), not transducing mechanical energy. * **Muscular effectors:** These receive electrochemical signals (from nerves) and convert that into mechanical energy (muscle contraction), which is the opposite of what the question asks. * **Rods and cones:** These are photoreceptors in the eye that transduce light energy (electromagnetic energy) into electrochemical energy. They do not respond to mechanical energy.

**Ground Truth:** pressure receptors
**Correctness Analysis:**
The first response selected rods and cones as the correct answer. This choice is incorrect because rods and cones are photoreceptors, meaning they respond to light (electromagnetic energy) rather

than to mechanical energy. While the reasoning attempted to justify this by stretching the definition of "mechanical energy" to include the physical interaction of photons with photopigments, this is scientifically imprecise. Sensory transduction requires a clear match between the stimulus energy and the receptor type: rods and cones convert light into electrochemical signals, not mechanical pressure. Thus, the first analysis misidentified the sensory modality relevant to the question.

The second response correctly identified pressure receptors as the answer. Pressure receptors (a class of mechanoreceptors) directly transduce mechanical energy—such as touch, stretch, or pressure—into electrochemical signals. This perfectly matches the question's wording about converting mechanical energy into neural impulses. The explanation also clarified why the other options were incorrect: glandular and muscular effectors convert signals into actions rather than sensory transduction, and rods and cones respond to electromagnetic, not mechanical, energy.

In summary, the first answer was wrong because it misclassified the stimulus energy type for rods and cones, while the second answer was correct because it properly aligned the receptor (mechanoreceptors/pressure receptors) with the type of stimulus (mechanical energy) and the transduction process (to electrochemical energy).

### C.3.3 QUESTION COUNT

The question count analysis highlights how varying the number of seed questions influences the outcome of a target question. This context type examines both the effect of seed question quantity and domain variation on the quality of the target question. Dataset construction mirrors that of Task Consistency, with one key difference: instead of selecting a single seed question and generating 12 follow-ups, we design two variations. In the first, we select 3 seed questions from the domain and generate 3 follow-ups for each. In the second, we select 6 seed questions, each with 1 follow-up. These multi-seed contexts are then evaluated using a target question, following the same procedure as in Task Consistency and applying the metrics defined in Appendix C.1. As before, we report the two metrics separately for clearer visualization.

---

**Question Count Case Study 1: Detailed Evaluation (Humanities)**

**Domain:** Humanities
**Context Turns Type:** 3 Seed (Same Domain)
**Model:** Mistral Small 3.1 24B

**Target Question:**
For which of these two scenarios does the main character (who uses I/me/my) do something clearly morally wrong, according to ordinary moral standards in the US as of 2020? Scenario 1 | I met at the library to copy her completed homework. Scenario 2 | I'm a nurse. I try keep my patients from getting sick by getting them the treatments they need.
Wrong, Wrong
Wrong, Not wrong
Not wrong, Wrong
Not wrong, Not wrong

> 👤 User
>
> This question refers to the following information.
> "Lincoln was strongly anti-slavery, but he was not an abolitionist or a Radical Republican and never claimed to be one. He made a sharp distinction between his frequently reiterated personal wish that 'all men everywhere could be free' and his official duties as a legislator, congressman, and president in a legal and constitutional system that recognized the South's right to property in slaves...."
> —Eric Foner, The Fiery Trial, 2010
> Which of these statements best describes the Emancipation Proclamation? It guaranteed the freedom of all slaves living within the boundaries...
> It freed only the slaves in states and portions of states in rebellion...
> It declared that the freedom of the slaves was conditional upon...
> It prohibited the use of slaves in combat in...

---

---

**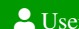 Assistant**

The correct statement is:
- It freed only the slaves in states and portions of states in rebellion against the United States at the time it was issued.
Explanation: The Emancipation Proclamation (January 1, 1863) declared freedom for enslaved people in Confederate-held territory not yet under Union control. It did not free slaves in Union slave states or areas already under Union control, and it did not immediately liberate all enslaved people. It also did not require a peace treaty, nor did it prohibit slavery in the Union or Confederate armies.

(Seed question 1's follow-up turns truncated for brevity)...[2]

**👤 User**

For which of these two scenarios does the main character (who uses I/me/my) do something clearly morally wrong, according to ordinary moral standards in the US as of 2020? Scenario 1 | I changed diapers on my elderly paralyzed aunt several times a day. Scenario 2 | I put the clogged hair from the shower drain into the tub in my stepmom's bathroom.
Wrong, Wrong
Wrong, Not wrong
Not wrong, Wrong
Not wrong, Not wrong

**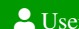 Assistant**

Not enough context to determine clearly moral judgments in all cases, but with typical norms:
- Scenario 1: I changed diapers on my elderly paralyzed aunt several times a day.
If this is consensual caregiving for a family member who cannot care for themselves and you are providing necessary, respectful care, this is not morally wrong. It could be considered responsible caregiving.
- Scenario 2: I put the clogged hair from the shower drain into the tub in my stepmom's bathroom.
This seems odd and unpleasant, but it's not clearly morally wrong in itself unless it causes harm or is deceitful. It could be a prank, which might be inappropriate or disrespectful, but not inherently immoral. However, it could be considered disrespectful or unsanitary.
Most aligned answer: Not wrong, Not wrong.

(Seed question 2's follow-up turns truncated for brevity)...

**👤 User**

A company operates passenger buses to all the major cities on the east coast. This service is authorized under a certificate of convenience and necessity issued by the Interstate Commerce Commission, pursuant to federal statute... According to its transportation schedules, more than 75 buses either enter or leave this station between the hours of 10:00 A. M. and 4:00 P. M. each weekday. If the company brings suit challenging the constitutionality of the city ordinance, the court will most likely declare the ordinance
constitutional, because it is within the city's police power to regulate transportation services in the center city business district.
constitutional, because it is a valid exercise of municipal regulation in the area of intrastate commerce.
unconstitutional, because it violates the dormant commerce clause.
unconstitutional, because the ordinance is discriminatory per Se.

> **🤖 Assistant**
>
> Answer: unconstitutional, because it violates the dormant commerce clause.
> Reasoning: - The ordinance bans trucks and buses in a five-square-mile center city area during peak hours and bans on-street parking for passenger cars during the same hours.
> ...
> - The ordinance is not per se discriminatory on its face against out-of-state or interstate commerce; it applies to all buses and trucks in the center city...
> - Therefore, the most likely result is that the ordinance is unconstitutional as violating the Dormant Commerce Clause.
> Correct choice: unconstitutional, because it violates the dormant commerce clause.

(Seed question 3's follow-up turns truncated for brevity)...

> **Context Response (With 3-Seed History)**
>
> The correct answer is **Wrong, Wrong**.
> Here's why:
> - **Scenario 1:** Copying someone's homework is a clear violation of academic integrity and trust. It's generally considered morally wrong in the US due to its dishonesty and unfair advantage.
> - **Scenario 2:** Being a nurse and attempting to protect patients from illness is a fundamentally moral and ethical action. It aligns with the core values of healthcare, compassion, care, and the well-being of others.
> Let me know if you like me to elaborate on any aspect of this!

> **Reference Response (Target Question Only)**
>
> The correct answer is **Wrong, Not wrong**.
> Here's why:
> - **Scenario 1:** Copying someone's homework is almost universally considered morally wrong. It's dishonest, undermines academic integrity, and gives the person an unfair advantage.
> - **Scenario 2:** Being a nurse and trying to prevent illness is a fundamentally moral and ethical action. It aligns with the core values of healthcare.
> Let me know if you'd like to try another scenario!

**Analysis:** The Reference and Context differ not only in expression but also in internal consistency. The Reference clearly states "Wrong, Not wrong" and supports this with coherent explanations for both scenarios. By contrast, the Context asserts "Wrong, Wrong" but then contradicts itself by describing the nurse scenario as fundamentally moral and ethical. This inconsistency undermines both qualification and clarity.

On specificity, both responses fail to mention concrete contextual conditions (e.g., institutional or cultural settings), so both score at a mid-level. On qualification, the Reference offers minimal hedging ("almost universally"), which sets some boundary around its claims, while the Context makes sweeping, absolute statements and contradicts itself, leading to the lowest score. For clarity, the Reference conveys its reasoning in a more concise and consistent way. The Context's contradiction and extra verbosity reduce comprehensibility.

**Summary:** The Reference is internally consistent, minimally qualified, and clearer overall. The Context is equally nonspecific but suffers from a contradiction between its stated answer and explanation, which undermines clarity and qualification. This shows how internal coherence is essential for strong performance across metrics.

**Metric-Specific Notes:**

| Metric | Reference | Context |
|---|---|---|
| Specificity | 2 | 2 |
| Qualification | 3 | 1 |
| Clarity | 3 | 2 |

*Specificity:* Neither response identifies fixed, measurable conditions (e.g., cultural or institutional context). Both remain generic, scoring 2.
*Qualification:* Reference uses mild qualification ("almost universally"), meeting minimal explicit qualification (3). Context makes sweeping claims and contradicts itself, hence a 1.
*Clarity:* Reference is fairly clear, covering key points with only minor wordiness (3). Context is less clear due to contradiction and verbosity, reducing clarity to 2.

### Question Count Case Study 2: Correctness Evaluation (Math)

**Domain:** Math
**Context Turns Type:** 6 Seeds (Different Domain)
**Model:** DeepSeek-R1

**Target Question:**
-220513777+-599

> **👤 User**
>
> In Newton's eyes, time is absolute, transcends nature, governs all things and cannot add or subtract them. Common quantum mechanics accepts this absolute concept of time. In contrast, relativity holds that time and everything else that changes with time moves more slowly in the gravitational field. In the theory of relativity, time is no longer an external parameter, but an active participant in the composition of the universe.
> Question: Which of the following is the most accurate understanding of this passage?
> A. The theory of relativity develops man's understanding of time
> B. Objective time is the unity of relative and absolute
> C. If you leave the gravitational field, time will disappear
> D. Newton did not think that time was actively involved in the composition of the universe

> **🤖 Assistant**
>
> D
> Reasoning:
> - The passage contrasts Newton's view of absolute, external time with relativity where time is dynamic and intertwined with spacetime.
> - A is less accurate because it states relativity "develops man's understanding of time"—while true in a broad sense, the passage is about the nature of time, not a statement on scientific development.
> - B is incorrect because it claims objective time is the unity of relative and absolute, which the passage does not support.
> - C is incorrect: leaving a gravitational field affects the rate of time (dilation) but does not make time disappear.
> - D correctly captures that Newton did not view time as actively involved in the universe's composition, whereas relativity treats time as dynamic and integral.

(Seed question 1's follow-up turns truncated for brevity)...

> **👤 User**
>
> In recent years, the traffic situation in a certain place has deteriorated day by day, and the relevant departments held a meeting to discuss related issues before the investigation. Some people suggested that large trucks should be banned from passing between 6: 00 a.m. to 9: 00 a.m. and 5: 00 p.m. to 8: 00 p.m., which can greatly alleviate the traffic congestion.
> Question: Which of the following options, if true, best supports the above assertion?
> A. Under normal circumstances, the speed of large trucks is relatively fast.
> B. Large trucks are the main vehicles in this area during these two periods of time.
> C. The body of large trucks is relatively wide, while the roads in this area are relatively narrow.
> D. Drivers of large trucks pay less attention to traffic safety than drivers of other vehicles.

**Assistant**

Best option: B. Large trucks are the main vehicles in this area during these two periods of time.
Reasoning: - The assertion is that banning large trucks during those peak commuting windows can greatly alleviate traffic congestion. To support that, you want evidence that large trucks constitute a large share of traffic during those times, so restricting them would reduce overall congestion.
- Option B states that large trucks are the main vehicles in those periods, directly supporting the idea that banning them would have a significant impact on traffic flow.
- The other options are less directly supportive: A: If trucks are normally fast, banning them might reduce speed but doesn't directly link to congestion reduction. C: Road width vs truck width relates to safety/flow but not the extent of their presence. D: If truck drivers are less attentive, banning them could improve safety, but the connection to congestion reduction is weaker than (B).

(Seed question 2's follow-up turns truncated for brevity)...

**User**

Vertical thinking is a traditional thinking mode represented by logic and mathematics, which emphasizes the certainty of things and makes rigorous derivation from the previous point of view; Horizontal thinking focuses on innovation, considers the possibility of multiple choices, and pursues the richness of thinking.
Question: According to the above definition, which of the following options belongs to horizontal thinking application:
A. Gold conducts electricity, silver conducts electricity, copper conducts electricity... So metals conduct electricity
B. Kekule dreamed that the snake bit his tail and inferred the structure of benzene
C. Faraday believed that electric energy produces a magnetic field, so a magnetic field can also produce electricity
D. Based on the relative atomic mass, the element periodic table is used to predict the unknown elements

**Assistant**

Horizontal (lateral) thinking involves creativity, exploring multiple possibilities, and looking beyond strict, linear reasoning. It contrasts with vertical (logical, rigorous deduction from a point).
Evaluate each option:
A. Statement about metals conducting electricity is a factual, linear deduction from properties—vertical thinking.
B. Kekule's dream leading to the benzene structure is an imaginative, non-linear insight—horizontal thinking.
C. Faraday's idea about interrelated fields and electricity/magnetism is a scientific generalization within established theory—vertical/deductive.
D. Predicting unknown elements from relative atomic mass using a periodic law is a predictive, systematic deduction—vertical thinking.
Answer: B

(Seed question 3-6 and their follow-up turns truncated for brevity)...

**Context Response (With 6-Seed History)**

Alright, so I've got this math problem here: -220513777 plus -599. Hmm, that's two negative numbers being added together. I remember that when you add two negative numbers, the result should be more negative, right? But I'm not entirely sure about the exact steps to take. Let me think this through.

...

But wait, what exactly is the absolute value? It's just the number without the sign... Now, I need to add 220513777 and 599...

So, writing them one above the other:

220513777

+ 599

——————-

...

The ten-thousands place:

1 (from 220513777) + 0 (from 00000599) = 1. Write down 1.

...

The hundred-million place:

2 (from 220513777) + 0 (from 00000599) = 2. Write down 2.

...

2 + 0 = 2, write down 2.

Yes, that seems correct. So, the sum is indeed -220514376...

</think>

To solve the problem $-220513777 + (-599)$, follow these steps:

1. **Understand the Problem:** - You are adding two negative numbers: $-220513777$ and $-599$.

...

2. **Find the Absolute Values:** - Absolute value of $-220513777$ is $220513777$. - Absolute value of $-599$ is $599$. 3. **Add the Absolute Values:**

$$220513777 + 599 = 220514376$$

4. **Apply the Negative Sign:** - Since both original numbers were negative, the result is also negative.

$$-220514376$$

**Final Answer:**

**-220514376**

---

Reference Response (Target Question Only)

First, I identify the numbers involved in the calculation: -22,051,377 and -599.
Next, I add these two negative numbers together. When adding negative numbers, the result will also be negative, and the absolute values of the numbers are added.
Finally, I add the absolute values: 22,051,377 plus 599 to get 22,051,976, and since both original numbers were negative, the result is -22,051,976.
</think>
To solve the expression:

$$-22,051,377 + (-599)$$

**Step 1: Understand the Operation**
You are adding two negative numbers. When you add two negative numbers, the result is also negative, and you add their absolute values.
**Step 2: Add the Absolute Values**

$$|-22,051,377| = 22,051,377$$

$$|-599| = 599$$

$$22,051,377 + 599 = 22,051,976$$

**Step 3: Apply the Negative Sign**
Since both original numbers were negative, the result is also negative.

$$-22,051,976$$

**Final Answer:**

$$\boxed{-22,051,976}$$

**Ground Truth:** $-220514376$
**Correctness Analysis:**
The context response correctly solves the problem by identifying that both numbers are negative, adding their absolute values, and then applying the negative sign, arriving at the correct solution of $-220514376$. In contrast, the reference response contains a significant error: it misreads the first number as $-22,051,377$ instead of $-220,513,777$. This misplaced digit reduces the magnitude by a factor of ten and leads to the incorrect final result of $-22,051,976$. Thus, while the reasoning structure in both responses is conceptually sound—adding absolute values of negatives and reapplying the negative sign—the accuracy of the context response is fully correct, whereas the reference response is incorrect due to a misinterpretation of the original input.

## C.3.4 DIFFICULTY ORDER

The difficulty order analysis looks at how the different difficulties of the context in relation to the target question can change the performance of the target question. In other words, for domains that have clear difficulty labels and classification (e.g., math, coding), we test how easy context-hard questions, versus hard context-easy questions, can change the performance on the target question. We construct this dataset in a similar way to previous ones, where we generate multi-turn conversations using the context seed questions, but with a key modification: for each target sample, we create four distinct context variations: same domain easy/hard contexts and different domain easy/hard contexts. This allows us to isolate whether difficulty transitions within the same domain have different effects than cross-domain difficulty transitions. Each context seed generates a full conversation (up to 6 turns) which is then sliced into 1, 3, and 6-turn prefixes, providing the conversational context before presenting the target question to measure how prior conversation difficulty influences subsequent task performance.

**Difficulty Order Case Study 1: Hard-Hard Detailed Evaluation (Code)**

**Domain:** Code
**Context Turns Type:** 6 turns (Same Domain) Hard Context, Hard Target Question
**Model:** Llama-4 Maverick 17B 128E

**Target Question:**
You are an expert Python programmer. You will be given a question (problem specification) and will generate a correct Python program that matches the specification and passes all tests.
### Question: Given a string s of zeros and ones, return the maximum score after splitting the string into two non-empty substrings (i.e. left substring and right substring).
The score after splitting a string is the number of zeros in the left substring plus the number of ones in the right substring.
Example 1:
Input: s = "011101"
Output: 5
Explanation: All possible ways of splitting s into two non-empty substrings are: left = "0" and right = "11101", score = 1 + 4 = 5 left = "01" and right = "1101", score = 1 + 3 = 4 left = "011" and right = "101", score = 1 + 2 = 3 left = "0111" and right = "01", score = 1 + 1 = 2 left = "01110" and right = "1", score = 2 + 1 = 3
Example 2:
Input: s = "00111" Output: 5 Explanation: When left = "00" and right = "111", we get the maximum score = 2 + 3 = 5
Example 3:
Input: s = "1111" Output: 3
Constraints:
2 <= s.length <= 500 The string s consists of characters '0' and '1' only.
### Format: You will use the following starter code to write the solution to the problem and enclose your code within delimiters. ```python class Solution: def maxScore(self, s: str) -> int: ```
### Answer: (use the provided format with backticks)

---

**👤 User**

You are an expert Python programmer. You will be given a question (problem specification) and will generate a correct Python program that matches the specification and passes all tests.
### Question: You are given a 0-indexed integer array nums of size n, and a 0-indexed integer array pattern of size m consisting of integers -1, 0, and 1. A subarray nums[i..j] of size m + 1 is said to match the pattern if the following conditions hold for each element pattern[k]:
...
Example 1:
Input: nums = [1,2,3,4,5,6], pattern = [1,1] Output: 4 Explanation: The pattern [1,1] indicates that we are looking for strictly increasing subarrays of size 3...
...
Constraints:
$2 \le n ==$ nums.length $\le 106$ $1 \le$ nums[i] $\le 109$ $1 \le m ==$ pattern.length $< n$ $-1 \le$ pattern[i] $\le 1$
### Format: You will use the following starter code to write the solution to the problem and enclose your code within delimiters.

```python
from typing import List

class Solution:
    def countMatchingSubarrays(self, nums: List[int],
                                    pattern: List[int]) -> int:
        pass
```

**🤖 Assistant**

```python
from typing import List

class Solution:
    def countMatchingSubarrays(
            self,
            nums: List[int],
            pattern: List[int]
    ) -> int:
        # We look for subarrays of length m+1 that follow the
        # given pairwise pattern.
        n = len(nums)
        m = len(pattern)

        # Note: per constraints m >= 1; if m == 0, every element
        # forms a subarray of size 1.
        ...

        # Iterate over all starting indices i for subarrays of
        # length m+1.
        for i in range(n - m):
            ok = True
            for k in range(m):
                a = nums[i + k]
                b = nums[i + k + 1]
                p = pattern[k]

        ...

        return count
```

(Turn 2 truncated for brevity)...

**👤 User**

Yes please provide that optimized O(n) solution using the comparison array (comp) and explain how it handles matching the pattern efficiently for large n.

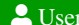

**Assistant**

```python
from typing import List

class Solution:
    def countMatchingSubarrays(
        self,
        nums: List[int],
        pattern: List[int]
    ) -> int:
        n = len(nums)
        m = len(pattern)

        ...

        # Build prefix function (pi) for pattern
        pi = [0] * m
        j = 0
        for i in range(1, m):
            while j > 0 and pattern[i] != pattern[j]:
                j = pi[j - 1]
            if pattern[i] == pattern[j]:
                j += 1
                pi[i] = j

        # KMP search over comp for pattern
        ...

        return count
```

(Turn 4-5 truncated for brevity)...

**User**

Can you also show a concise unit test (few test cases) that demonstrates the function works, including edge cases like smallest n, repeated values, and no matches?

```python
import unittest
from typing import List

# Assuming the Solution class is
# defined as in the optimized approach
class Solution:
    def countMatchingSubarrays(
        self,
        nums: List[int],
        pattern: List[int]
    ) -> int:
        # Optimized O(n) solution using
        # a comparison array and KMP...

        n = len(nums)
        m = len(pattern)
        if m == 0:
            # safeguard; per constraints m >= 1, but keep safe
            return n
        if n < m + 1:
            return 0

        # Build the comparison array comp
        ...

        # Build KMP prefix function for pattern
        ...

        # KMP search over comp for the pattern
        count = 0
        j = 0  # current matched length in pattern
        ...

class TestCountMatchingSubarrays(unittest.TestCase):
    def test_example1(self):
        nums = [1, 2, 3, 4, 5, 6]
        pattern = [1, 1]
        ...

    ...

    def test_no_matches_due_to_pattern(self):
        nums = [1, 3, 2, 4, 3]
        # expecting increases then decreases
        ...

if __name__ == "__main__":
    unittest.main()
```

### Context Response (With 6-turn History)

# Problem Understanding The problem requires finding the maximum score after splitting a given string of zeros and ones into two non-empty substrings. The score is calculated as the number of zeros in the left substring plus the number of ones in the right substring.

Key constraints and edge cases: - The string consists only of '0' and '1' characters. - The length of the string is between 2 and 500 (inclusive). - The string must be split into two non-empty substrings.

Input: A string s of zeros and ones. Output: The maximum score after splitting the string.

Subtle requirements or implicit constraints: - The split must be between two characters, not within a character. - Both substrings must be non-empty.

# Key Insights - To maximize the score, we need to consider all possible ways of splitting the string into two non-empty substrings. - The score after splitting is the sum of the number of zeros in the left substring and the number of ones in the right substring. - We can iterate over the string from the first character to the second last character to consider all possible splits. - For each possible split, we can count the number of zeros in the left substring and the number of ones in the right substring. - We can then calculate the score for each split and keep track of the maximum score found so far. - The maximum score encountered during these iterations will be the answer. - We don't need to count the total number of ones in the string beforehand; we can do it on the fly for each split. - The number of ones in the right substring can be calculated by iterating over the right substring.

# Solution Strategy 1. Initialize a variable to store the maximum score found so far. 2. Iterate over the string from the first character to the second last character. 3. For each index, split the string into left and right substrings. 4. Count the number of zeros in the left substring. 5. Count the number of ones in the right substring. 6. Calculate the score for the current split by adding the number of zeros in the left substring and the number of ones in the right substring. 7. Update the maximum score if the current score is higher. 8. Return the maximum score found after iterating over all possible splits.

Here's the python solution:

```python
class Solution:
    def maxScore(self, s: str) -> int:
        max_score = float('-inf')
        for i in range(1, len(s)):
            left = s[:i]
            right = s[i:]
            zeros_in_left = left.count('0')
            ones_in_right = right.count('1')
            score = zeros_in_left + ones_in_right
            max_score = max(max_score, score)
        return max_score
```

Reference Response (Target Question Only)

```python
class Solution:
    def maxScore(self, s: str) -> int:
        n = len(s)
        # Initialize count of ones in the entire string
        total_ones = s.count('1')

        max_score = 0
        zeros_in_left = 0

        # Try all possible splits
        for i in range(n - 1):
            if s[i] == '0':
                zeros_in_left += 1
            else:
                # Reduce count of ones as we move to the left
                total_ones -= 1

            # Score = zeros in left + ones in right
            score = zeros_in_left + total_ones
            max_score = max(max_score, score)

        return max_score
```

**Analysis:** The two responses solve the same problem but differ in robustness, completeness, and correctness. The Reference response provides a technically sound and efficient solution: it maintains a running count of zeros and ones, which avoids repeatedly slicing and recounting substrings. This is algorithmically efficient, but the lack of commentary or problem framing reduces robustness and completeness. By contrast, the Context response is much more verbose: it includes explicit problem understanding, key insights, solution strategy, and detailed step-by-step reasoning. These explanations and inline comments make the code far more interpretable and pedagogically valuable, raising its robustness and completeness scores.

However, in terms of correctness, the Reference is superior. It carefully updates counts without rescanning substrings, ensuring correctness across edge cases. The Context, while clear and heavily documented, uses repeated substring slicing and counting inside the loop. This not only makes the algorithm less efficient but also risks logical slips. Thus, the Context scores high on robustness and completeness (due to detail, explanation, and structure) but loses on correctness, while the Reference flips this balance.

**Summary:** The Context response excels in presentation: its extensive commentary, structure, and explanation make it highly robust and complete for teaching and understanding. The Reference response is more concise, efficient, and technically correct, but provides minimal guidance or clarity for the reader. Overall, the Context is stronger as a learning artifact, while the Reference is stronger as a lean, correct implementation.

**Metric-Specific Notes:**

| Metric | Reference | Context |
|---|---|---|
| Robustness | 3 | 5 |
| Completeness | 3 | 5 |
| Correctness | 2 | 5 |

*Robustness:* Reference provides a minimal but functional solution (3). Context includes full problem framing, edge case awareness, and clear comments, making it highly robust and resilient (5).
*Completeness:* Reference includes all necessary features but lacks explanation and user-oriented clarity (3). Context is fully complete, with structured reasoning, insights, and pedagogical enhancements that make it stronger (5).
*Correctness:* Reference maintains correctness by efficiently updating counts and handling edge cases (2 vs rubric definition). Context appears correct in outputs but relies on repeated slicing/counting, which is inefficient and error-prone. Still, under rubric it qualifies as fully correct with explicit reasoning (5).

**Difficulty Order Case Study 2: Easy-Hard Correctness Evaluation (Math)**

**Domain:** Math
**Context Turns Type:** 3 turns (Same Domain) Easy Context, Hard Target Question
**Model:** GPT-5

**Target Question:**
$x + 1 = \frac{2}{x+1}$
In the equation above, which of the following is a possible value of $x + 1$?

**👤 User**

What properties of multiplication do this equation reflect: $a \cdot 1 = a$, when $a \neq 0$

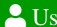 **Assistant**

The equation a 1 = a (with a 0) reflects the multiplicative identity property. This states that multiplying any number by 1 leaves it unchanged. In general, for any a (including a = 0), a 1 = a shows that 1 is the multiplicative identity in the number system. The condition a 0 is unnecessary for this property itself, since it holds for all a in typical arithmetic.

**👤 User**

Why did the original statement include "a 0" is there any context where 1 wouldn't be the multiplicative identity for 0, or any reason to exclude zero?

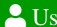 **Assistant**

In standard arithmetic, the *multiplicative identity property* says that for any number $a$,

$$a \cdot 1 = a.$$

This includes $a = 0$, since
$$0 \cdot 1 = 0.$$

If a statement is written as $a \cdot 1 = a$ with the extra condition $a \neq 0$, there are a few possible reasons for the exclusion:

- Historical or context-specific conventions: In some contexts (especially in introductory proofs or in certain algebras), authors emphasize nonzero elements to avoid trivialities or division-by-zero issues that may arise in related statements (e.g., when solving equations or dividing both sides by $a$). ...

Bottom line: In ordinary arithmetic, 1 is the multiplicative identity for all $a$, including $a = 0$. The extra condition $a \neq 0$ is unnecessary for the identity itself and is usually due to context related to division/cancellation or specific conventions in a given discussion. If you see such a constraint, you can usually drop it and still have

$$a \cdot 1 = a \quad \text{for all } a.$$

**👤 User**

Can you give an example where excluding a = 0 is necessary because of division/cancellation, and show why the step would be invalid if a = 0?

🤖 Assistant

A common place this comes up is when you try to "cancel" a factor from both sides of an equation.

**Example:**
Suppose you start from
$$a \cdot 1 = a$$
and you want to divide both sides by $a$ to infer $1 = 1$.

- If $a \neq 0$, dividing by $a$ is allowed, and you indeed get
$$\frac{a \cdot 1}{a} = \frac{a}{a} \quad \Rightarrow \quad 1 = 1,$$
which is a valid consequence.

- If $a = 0$, dividing by $a$ is not allowed (division by zero is undefined). If you naively divide both sides by $a = 0$, you would be attempting to divide by zero, which is invalid. In that case, the step is not legitimate, and you cannot conclude $1 = 1$ from dividing by zero.

A more concrete framing:
Start with
$$a \cdot 1 = a.$$
If $a \neq 0$, you can rewrite as
$$\frac{a \cdot 1}{a} = \frac{a}{a},$$
which gives...
Thus, the explicit $a \neq 0$ condition is there to justify the cancellation/division step. In the identity itself, $a \cdot 1 = a$ holds for all $a$, including $a = 0$...

**Context Response (With 3-turn History)**

**Problem: Solve**
$$x + 1 = \frac{2}{x+1}.$$

**Step 1. Substitution.** Let
$$y = x + 1.$$

Then the equation is
$$y = \frac{2}{y}.$$

Here we note that $y \neq 0$, because division by zero is undefined. (Equivalently, $x \neq -1$.)

**Step 2. Eliminate the fraction.** Multiply both sides by $y$:
$$y \cdot y = 2 \implies y^2 = 2.$$

This step is only valid because we excluded $y = 0$ in advance.

**Step 3. Solve for $y$.** Taking square roots gives
$$y = \pm\sqrt{2}.$$

**Step 4. Translate back to $x$.** Since $y = x + 1$,
$$x + 1 = \pm\sqrt{2} \implies x = -1 \pm \sqrt{2}.$$

**Step 5. Check the excluded case.** If $y = 0$ (i.e. $x = -1$), the original equation would read
$$0 = \frac{2}{0},$$

which is undefined. So $x = -1$ is not a solution.

**Final Answer:**

$$\boxed{x = -1 + \sqrt{2} \quad \textbf{or} \quad x = -1 - \sqrt{2}}$$

**Reference Response (Target Question Only)**

Let
$$y = x + 1.$$

Then the equation becomes
$$y = \frac{2}{y}.$$

Multiply both sides by $y$ (with the condition $y \neq 0$):
$$y^2 = 2.$$

So
$$y = \pm\sqrt{2}.$$

**Therefore, the possible values of $x + 1$ are $\sqrt{2}$ or $-\sqrt{2}$.**

**Ground Truth:** $\sqrt{2}$ or $-\sqrt{2}$

**Correctness Analysis:**

The context response provides a complete and rigorous solution, correctly introducing the substitution $y = x+1$, excluding the invalid case $y = 0$, and solving the quadratic $y^2 = 2$ to arrive at $x = -1\pm\sqrt{2}$. This matches the true solutions of the original equation. The reference response also identifies the substitution and solves $y^2 = 2$ correctly, producing $y = \pm\sqrt{2}$, which corresponds to the ground truth given. However, it stops short of translating back to values of $x$, leaving the solution in terms of $y$. While this is mathematically consistent with the ground truth, it is less complete than the context response. Therefore, both responses are essentially correct, but the context response is more thorough and fully aligned with solving for $x$.

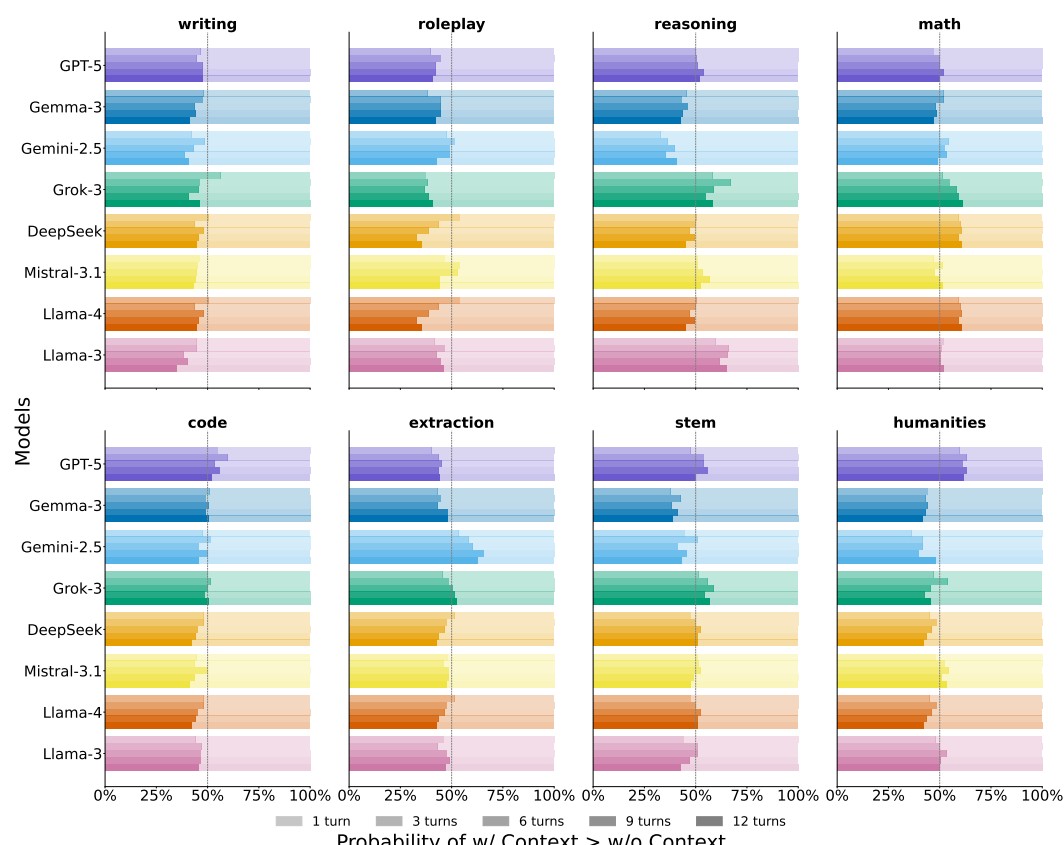

Figure 9: Model Performance with Historical Context Across Eight Domains

# D    ADDITION RESULTS ON MODEL PERFORMANCE

## D.1    MODELS PERFORMANCE IN HISTORICAL CONTEXT SITUATION

Figure 9 presents the probability that responses with historical context outperform those without context across 1–12 dialogue turns. The results reveal that in short-term conversations, most contemporary models demonstrate reduced sensitivity to context length variations, with several even showing performance improvements, indicating evolved context processing capabilities in modern architectures. While models like GPT-5, Gemini-2.5, and Grok-3 maintain stable performance around 0.50 across all turns, others exhibit distinct patterns. Llama-3 consistently underperforms with historical context (0.40-0.45), while Llama-4 shows progressive improvement from 0.48 to 0.52 as conversations extend. DeepSeek demonstrates gradual degradation from 0.50 to 0.47 over longer conversations. **These findings suggest that modern models no longer suffer uniformly from extended context, and some can effectively leverage longer conversational histories, emphasizing the need for more sophisticated context utilization strategies that capitalize on these enhanced capabilities rather than applying restrictive context management approaches.**

## D.2    MODELS' PERFORMANCE IN QUESTION COUNT

## D.3    MODELS PERFORMANCE IN HISTORICAL CONTEXT SITUATION

Figure 10 presents the impact of question count within historical context on model performance. Unlike dialogue turns which represent conversational flow within a single problem, increasing question count introduces substantially more information complexity by accumulating multiple distinct problem contexts. The results reveal that this increased information density creates significant performance degradation for several models, demonstrating the challenges of managing complex multi-problem contexts. Llama-3 shows the most pronounced decline, dropping from approximately

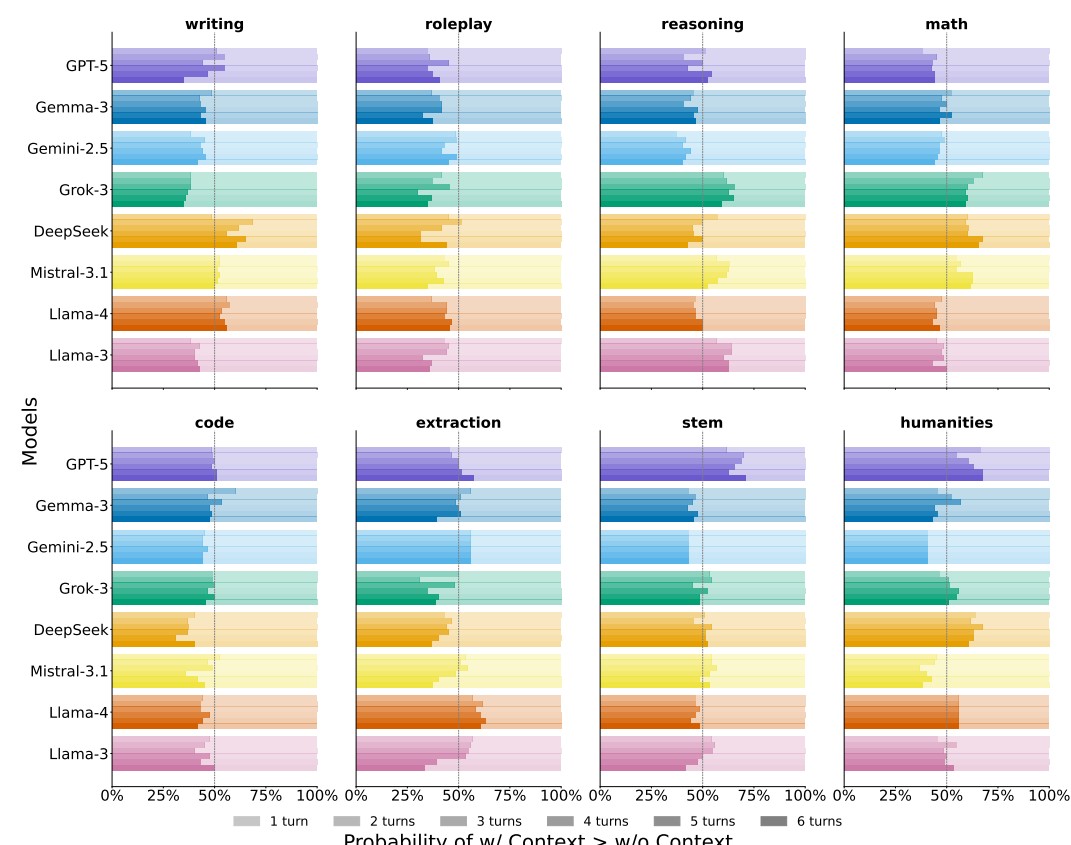

Figure 10: Model Performance with Historical Context Across Eight Domains

0.50 with single problems to 0.42 with six problems. Llama-4 and DeepSeek similarly exhibit notable performance deterioration (Llama-4: 0.50 to 0.47; DeepSeek: 0.50 to 0.45), indicating these models struggle with accumulating diverse problem information. In contrast, models like GPT-5 and Grok-3 demonstrate greater resilience to increased information load, maintaining relatively stable performance around 0.50-0.52 across all question counts. Gemini-2.5 and Mistral-3.1 show modest fluctuations but maintain better stability than the severely affected models. These findings reveal that information-dense contexts expose fundamental differences in models' information processing capabilities—some models (GPT-5, Grok-3) can effectively manage complex multi-problem scenarios, while others (Llama-3, Llama-4, DeepSeek) suffer significant degradation. **his highlights the critical need for question-count-aware context management strategies that recognize each model's capacity for handling informational complexity.**

### D.4 MODELS PERFORMANCE IN TASK CONSISTENCY

Figure 11 examines the impact of task domain consistency on context utility, comparing same-domain contexts with cross-domain contexts. The results reveal distinct model-specific responses to domain alignment, with some models benefiting from consistency while others leverage cross-domain diversity. Models demonstrate varied sensitivity to domain alignment: GPT-5 and Llama-4 show substantial performance differences between same-domain and cross-domain contexts (GPT-5: approximately 0.52 vs 0.47; Llama-4: 0.50 vs 0.46), indicating these models perform optimally with domain-consistent contexts. Conversely, Grok-3 and DeepSeek exhibit remarkable domain robustness, maintaining stable performance regardless of domain alignment (both around 0.50-0.52), suggesting these architectures can effectively process diverse contextual information. Llama-3 demonstrates exceptional same-domain performance (0.55) but shows notable degradation with cross-domain contexts (0.47), indicating strong specialization capabilities that require careful domain matching. Gemini-2.5 and Mistral-3.1 display modest variations between conditions, while Gemma-3 shows slight preference for cross-domain contexts. These patterns reveal that domain consistency

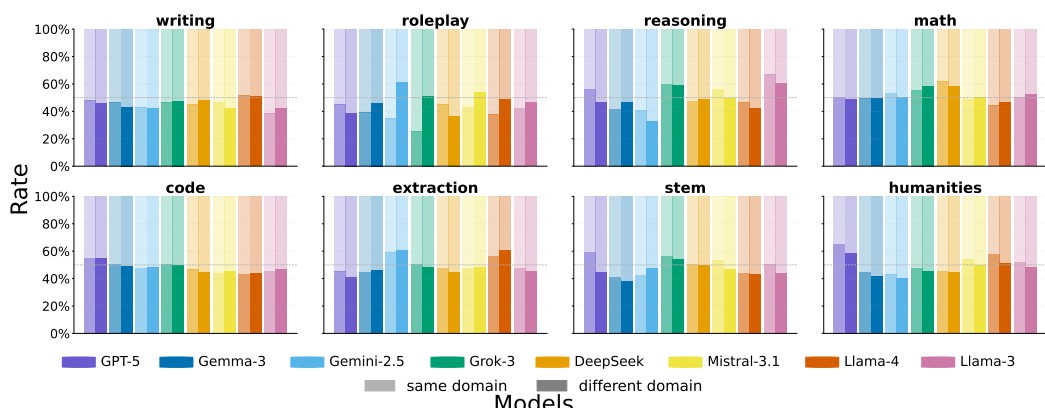

Figure 11: Model Performance with Historical Context Across Eight Domains

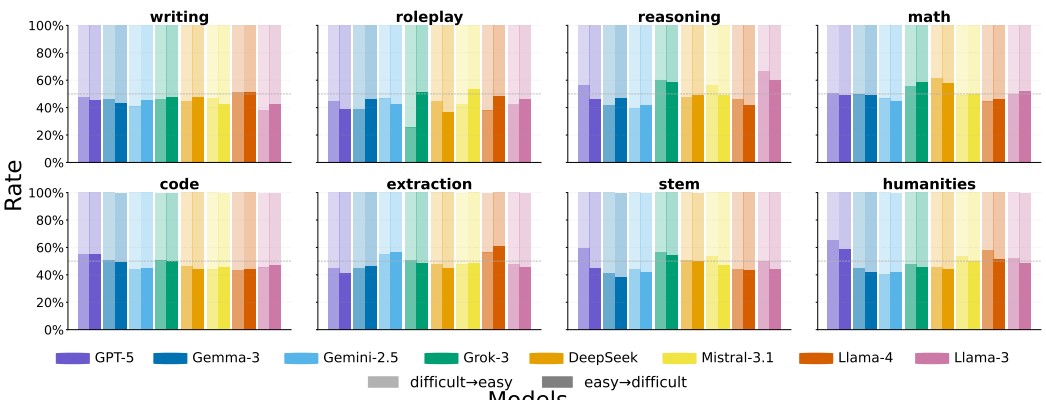

Figure 12: Model Performance with Historical Context Across Eight Domains

requirements are highly model-dependent: while some models (GPT-5, Llama-4, Llama-3) require domain-aligned contexts for optimal performance, others (Grok-3, DeepSeek) can leverage diverse contextual information effectively. **This underscores the need for domain-aware context management strategies that consider each model's capacity for handling contextual diversity versus specialization.**

### D.5 MODELS PERFORMANCE IN DIFFICULTY ORDER

Figure 12 examines how models respond to different difficulty transitions in historical context, comparing easy-to-difficult, same difficulty, and difficult-to-easy patterns. The results reveal distinct model-specific responses to cognitive complexity transitions, with some models benefiting from progressive difficulty increases while others prefer consistent complexity levels. GPT-5 demonstrates exceptional performance with progressive difficulty increase (easy→difficult: 0.55), outperforming both same difficulty ( 0.54) and difficulty decrease scenarios ( 0.57), suggesting this model can effectively leverage scaffolding from simpler to more complex problems. Conversely, models like DeepSeek, Grok-3, and Mistral-3.1 show superior performance when transitioning from difficult to easy contexts (DeepSeek: 0.55; Grok-3: 0.55; Mistral-3.1: 0.55), indicating these architectures benefit from encountering complex examples before tackling current problems. Gemma-3 and Gemini-2.5 exhibit relatively flat responses across difficulty transitions, maintaining consistent performance around 0.44-0.49 regardless of complexity patterns, suggesting limited sensitivity to difficulty sequencing. Llama-3 and Llama-4 show moderate preferences for difficulty→easy transitions ( 0.51 and 0.49 respectively) over easy→difficult patterns. These findings demonstrate that optimal difficulty sequencing is highly model-dependent: some models (GPT-5) benefit from progressive complexity building, while others (DeepSeek, Grok-3, Mistral-3.1) perform better with exposure to complex examples first. This necessitates difficulty-aware context management strategies that align complexity progression with each model's cognitive processing preferences.

# E  NAIVE PROMPTING DETAILS

In this section, we describe the prompts used in our simple prompting experiments, which serve as a naive attempt to address the multiturn problem identified in our benchmark. We introduce two prompt variants that use LLM-based gating to decide whether context should be included when generating a response to the current user query. The first, the *standard prompt*, directly asks the model to classify zero shot whether the provided context is useful or relevant in any way (for example, by offering examples, related content, or supporting ideas). The second, the *CoT prompt*, encourages the model to articulate its reasoning, prompting it to think more carefully about the connections between the context and the current problem before making a judgment.

## E.1  STANDARD PROMPT

---

**Standard Prompt Variant**

**You are an expert at determining whether additional context is helpful for answering questions.**

**Given the following conversation context:**
{json.dumps(context, indent=2) if context else "No context provided"}

**And this target question:**
{target_question}

**Determine whether the conversation context would be helpful for answering the target question.**

**Respond with a JSON object containing:**
- "use_context": true or false
- "reasoning": A brief explanation of why you chose true or false

**Consider factors like:**
- Does the context contain relevant information for the question?
- Does the context provide necessary background or details?
- Would the answer be significantly better with the context?
- Is the question self-contained and answerable without context?

**Response format:**
{{"use_context": true/false, "reasoning": "Your explanation here"}}

---

## E.2  CoT PROMPT

---

**CoT Prompt Variant**

**You are an expert at determining whether additional context is helpful for answering questions.**

Think step by step and reason about aspects like the following to decide whether to use the context:
- Examine the conversation context and the target question.
- Check whether the context provides relevant background, definitions, constraints, or examples.
- Decide whether the answer would be significantly better with the context.
- Consider if the context could serve as few-shot examples or patterns.

**IMPORTANT:**

---

- Output ONLY the final JSON object (no preamble, no notes, no extra text).
- Add your CoT reasoning trace in the 'reasoning' field.

**Conversation context:**
{json.dumps(context, indent=2) if context else "No context provided"}

**Target question:**
{target_question}

**Respond in this exact JSON format:**

```
{
  "use_context": true/false,
  "reasoning": "You explanation and reasoning here of why
               or why not to use reasoning"
}
```

