# OpenReview forum: "Irrelevant Context Helps: Understanding the Impact of Context in Large Language Models"
_ICLR.cc/2026/Conference — ICLR 2026 Conference Withdrawn Submission_

### Official Review · Reviewer_nBbP · 2025-10-31

**Soundness:** 2
**Presentation:** 3
**Contribution:** 2
**Rating:** 4
**Confidence:** 3

**Summary:**

The paper challenges the common assumption that question-irrelevant dialogue history always harms LLMs. It introduces When2Read, a 31,900-pair benchmark spanning eight domains and systematically varying context factors (task consistency, multi-turn depth, question count, difficulty order) to compare answers with vs. without such history.

**Strengths:**

- The paper makes an explicit, useful distinction between relevant and question-irrelevant historical context and shows that the latter can sometimes act like a cognitive activator rather than mere noise. This is a nontrivial correction to the “filter-all-irrelevant” paradigm in long-context LLM systems.
- When2Read is large (31,900 pairs), covers eight realistic domains (writing, roleplay, reasoning, math, coding, extraction, STEM, humanities), and varies four contextual factors independently, which makes it possible to attribute gains/losses to specific context structures rather than to “long context” in general.
- The paper evaluates eight families (GPT-5, Llama-3/4, Gemini, DeepSeek, Grok-3, Gemma, Mistral) and shows that newer models are better at exploiting irrelevant context, while smaller/older ones still degrade—so the finding is not a quirk of one model.
- The output-length gate is model-agnostic, training-free, and outperforms standard prompting/CoT in many domains, which makes the paper useful for practitioners.

**Weaknesses:**

- Many pairwise quality decisions (when correctness ties) rely on GPT-5 rubric scoring. Although the authors report decent agreement with humans, this still risks evaluator bias and self-preference.
- The benchmark constructs “irrelevant” histories by mixing domains, depth, difficulty orders, and question counts, but the paper does not fully rule out latent topical/skill leakage (e.g., general math or explanation style bleeding across turns). A more formal irrelevance test (lexical/semantic overlap thresholds, task-type disjointness) would clarify how strictly “irrelevant” the contexts really are.
- The four context factors are explored over specific, handpicked sets (depth up to 12; question count up to 6; two difficulty orders). It is not fully clear that the main conclusions would survive deeper, denser, or messier real-world histories (e.g., 30+ mixed-topic customer turns). Sensitivity plots are suggestive but not exhaustive.

**Questions:**

- During data creation, how is “question-irrelevant” verified beyond domain mixing?
- The four context factors (task consistency, multi-turn depth, question count, difficulty order) are fixed to discrete sets. Why these grids, and how sensitive are findings to alternative values (e.g., deeper than 12 turns, >6 seed questions)?

---

### Official Review · Reviewer_sD3P · 2025-10-31

**Soundness:** 2
**Presentation:** 2
**Contribution:** 2
**Rating:** 2
**Confidence:** 4

**Summary:**

This paper introduces When2Read, a benchmark for studying the effects of question-irrelevant context on LLMs in multi-turn dialogues. Experiments on eight models show such context can occasionally help reasoning but often harms creative tasks. The authors propose a simple length-based filtering strategy to decide when to use context, reporting notable accuracy gains without additional training.

**Strengths:**

1. The paper tackles an underexplored question—whether question-irrelevant context can sometimes benefit LLM performance—challenging a widely held assumption in the field. The benchmark design systematically manipulates multiple context dimensions, offering a novel framework for controlled evaluation in multi-turn dialogue settings.

2. The dataset is large-scale (31,900 paired samples) and covers eight diverse task domains, providing a substantial empirical basis for analysis. The proposed length-based filtering strategy is simple, model-agnostic, and applicable to both open- and closed-source models without additional training.

3. The paper is generally well-structured, with clear descriptions of benchmark construction, experimental setup, and evaluation metrics. Figures and tables are easy to interpret, aiding comprehension of the results.

**Weaknesses:**

1. While human cognitive science is referenced, no predictive framework is developed for LLMs to explain when and why irrelevant context might help, weakening the theoretical grounding of the work.

2. All dialogues in the benchmark are generated by GPT-5 and evaluated by GPT-5, which introduces both generation and evaluation bias, and human verification covers less than 1.1% of samples, making it insufficient to rule out systematic bias.

3. Figure 4 shows that most model-task combinations degrade with irrelevant context, yet the narrative focuses on rare positive cases without statistical significance testing, which risks misleading conclusions.

4. The paper does not investigate why certain models or tasks benefit from irrelevant context, leaving the findings purely observational and limiting their practical applicability.

5. In table 3, the proposed method requires two inferences per query, whereas baselines require only one, giving it an unequal inference budget that may inflate performance gains and should be controlled for in comparisons.

6. The most relevant and widely used approach—semantic filtering of irrelevant context —is acknowledged in Related Work but omitted from experiments, and including it would provide a fairer and more complete evaluation.

7. In Fig.7, when considering only cases where output length increases, performance is not necessarily better than without context, indicating that length increase is not a reliable indicator of improvement; combined with Fig.4’s evidence of widespread performance drops, the occasional gains do not substantiate the paper’s central claim and fail to provide new insights.

**Questions:**

1. Why was the standard semantic filtering of irrelevant context, which is widely used and acknowledged in your Related Work, omitted from the experimental baselines? Including it could significantly strengthen the validity of your comparisons.

2. How would the proposed length-based filtering method perform if all methods were constrained to the same inference budget, i.e., only one inference per query?

3. Can you provide statistical significance tests for the improvements reported in Figure 4 and Figure 7, to confirm whether the observed gains are reliable rather than due to random variation?

4. What mechanisms or factors explain the few positive cases where irrelevant context improves performance? Are these linked to specific model architectures, task types, or context structures?

5. How do you mitigate bias introduced by using GPT-5 for both benchmark generation and evaluation, and what evidence can you provide that this bias does not materially affect your conclusions?

6. Given that Figure 7 shows length increase is not a reliable indicator of performance improvement, how do you justify using output length as the primary selection criterion?

7. Based on the evidence in Figure 4 and Figure 7, most cases show performance drops with irrelevant context, and even when output length increases, performance is not consistently better. Given this, how can readers or practitioners effectively leverage irrelevant context to improve model performance, as the paper claims? Please clarify the specific conditions, mechanisms, or guidelines under which irrelevant context is genuinely beneficial.

---

### Official Review · Reviewer_WKuT · 2025-11-01

**Soundness:** 2
**Presentation:** 3
**Contribution:** 3
**Rating:** 4
**Confidence:** 3

**Summary:**

This paper challenges the assumption that question-irrelevant context is universally harmful to LLMs. Using its new When2Read benchmark, the study shows this context has a bidirectional effect, sometimes improving performance significantly and other times causing degradation. Critically, it proposes a practical, training-free heuristic for adaptive context management: a response shorter than the no-context baseline strongly indicates poor performance. This work offers a valuable criterion for strategically leveraging conversational history in dialogue systems without requiring model retraining.

**Strengths:**

1. The paper's primary strength is its systematic investigation into the underexplored, nuanced effects of question irrelevant context. It introduces When2Read, the first large-scale, controlled benchmark for this purpose, providing a valuable resource for the community. This allows the work to move beyond simple assumptions and establish the highly situation-dependent nature of contextual utility with strong empirical evidence.
2. The finding that shorter outputs correlate with performance degradation is an elegant, training-free signal. This provides an immediately applicable solution for improving model performance without costly retraining, adding significant practical value to the paper's theoretical insights.

**Weaknesses:**

1. The analysis of the underlying mechanisms is insufficient. The paper does not delve into what is happening internally within the LLMs (e.g., altered attention patterns, a smoothed output distribution, or stylistic/structural priming) to produce these performance gains. Consequently, the fundamental questions remain unanswered: Why is the impact of irrelevant context bidirectional, and what is the root cause of this behavior?
2. The heuristic method increases the cost of practical application. Requiring two inferences for each query (one with context and one without) effectively doubles the inference cost and latency, making it impractical for real-world scenarios.
3. The paper heavily relies on GPT-5 as the judge for evaluating quality dimensions that are not objectively correct. While the authors present a consistency validation against human annotators, a potential conflict arises because GPT-5 is also a subject of the evaluation. This circular evaluation framework risks introducing a bias where the model may unfairly favor outputs that align with its own generation style. The experimental results should include a more diverse set of models as judges to mitigate this concern.
4. The study's design only tests a simple [history] + [query] format. A more granular analysis could be conducted by manipulating the insertion points of irrelevant context. For example, by varying its proximity to the relevant query or by interleaving relevant and irrelevant turns, one could test whether the impact is primarily driven by the most immediate irrelevant fragments.

**Questions:**

please see Weaknesses

---

### Official Review · Reviewer_a35L · 2025-11-12

**Soundness:** 2
**Presentation:** 3
**Contribution:** 2
**Rating:** 2
**Confidence:** 4

**Summary:**

This paper investigates the effect of prepending irrelevant contextual information to an LLM-based question answering task.  It constructs a new benchmark that is largely formed from existing datasets – the key contribution is that question-irrelevant content is added, and importantly, characterised in terms of task consistency, turn depth, turn breadth, and difficulty.  (These labels are judged by LLMs).

Several different tasks are included in the benchmark, and multiple SOTA LLMs are evaluated.  The findings are not very consistent across LLMs or conditions, but do reveal that sometimes, performance is higher when irrelevant context is included.  This is perhaps surprising in itself – but it does mean that the title should more accurately be "Irrelevant context SOMETIMES helps".

**Strengths:**

The paper is thorough and well-written.  There is a broad analysis, and it is informative to see the results of the various LLMs across different tasks.

I liked the idea of analysing the context across differing dimensions.

**Weaknesses:**

The results showed little consistency, with variation across tasks, LLMs and question types.  Other than the headline finding that irrelevant context _sometimes_ helps (which is perhaps somewhat surprising) there is not a clear take-home message from the paper.  The analysis is purely empirical and somewhat shallow.  Whilst I enjoyed the formulation in terms of utility, otherwise there was little attempt to dig deeper into the role of the context from a more theoretical perspective.

I suspect that that the context may be a proxy for some other hyper parameters that affect the style of LLM output.  You should have investigate this further, rather than suggested (in a somewhat vague way) that it analogous to "humans draw[ing] inspiration from seemingly irrelevant experiences.

It is true that LLMs are largely black boxes, but it is a bit unsatisfying that no explanations were given for the highly inconsistent performance across models and conditions.  Perhaps you could have developed or fine-tuned more controlled LLMs that would have enabled a deeper investigation.

**Questions:**

I was not convinced by the demonstration that longer answers achieving better scores is connected by the role of irrelevant context – what if you had chosen some other way to generate longer answers?  Perhaps you would have found the same effect, in the case that the metrics you used tend to have an inherent preference for longer answers.

---

### Author Response · Authors · 2025-12-02

We thank the reviewers for their thoughtful and constructive feedback. After careful consideration, we have decided to withdraw the submission in order to conduct further revisions and strengthen the work.

---

### Note · Authors · 2025-12-02

**Comment:**

We thank the reviewers for their thoughtful and constructive feedback. After careful consideration, we have decided to withdraw the submission in order to conduct further revisions and strengthen the work.

**Withdrawal Confirmation:**

I have read and agree with the venue's withdrawal policy on behalf of myself and my co-authors.